# Light-triggered and phosphorylation-dependent 14-3-3 association with NON-PHOTOTROPIC HYPOCOTYL 3 is required for hypocotyl phototropism

Lea Reuter[1,3], Tanja Schmidt [1,3], Prabha Manishankar[1], Christian Throm [1], Jutta Keicher[1], Andrea Bock[1], Irina Droste-Borel[2] & Claudia Oecking [1✉]

NON-PHOTOTROPIC HYPOCOTYL 3 (NPH3) is a key component of the auxin-dependent plant phototropic growth response. We report that NPH3 directly binds polyacidic phospholipids, required for plasma membrane association in darkness. We further demonstrate that blue light induces an immediate phosphorylation of a C-terminal 14-3-3 binding motif in NPH3. Subsequent association of 14-3-3 proteins is causal for the light-induced release of NPH3 from the membrane and accompanied by NPH3 dephosphorylation. In the cytosol, NPH3 dynamically transitions into membraneless condensate-like structures. The dephosphorylated state of the 14-3-3 binding site and NPH3 membrane recruitment are recoverable in darkness. NPH3 variants that constitutively localize either to the membrane or to condensates are non-functional, revealing a fundamental role of the 14-3-3 mediated dynamic change in NPH3 localization for auxin-dependent phototropism. This regulatory mechanism might be of general nature, given that several members of the NPH3-like family interact with 14-3-3 via a C-terminal motif.

[1] Center for Plant Molecular Biology (ZMBP), Plant Physiology, University of Tübingen, Tübingen, Germany. [2] Proteome Center Tübingen, University of Tübingen, Tübingen, Germany. [3] These authors contributed equally: Lea Reuter, Tanja Schmidt. ✉email: claudia.oecking@zmbp.uni-tuebingen.de

Developmental plasticity of plants is impressively demonstrated by the phototropic response, through which plants align their growth with incoming blue light (BL)[1]. Shoots typically grow towards the light by generating a lateral gradient of the growth-promoting phytohormone auxin. Here, the hormone concentration is higher on the shaded side as compared with the lit side, resulting in differential growth. It is well established that the phototropins phot1 and phot2 function as primary photoreceptors controlling phototropism in *Arabidopsis*[2–4]. Phototropins are plasma membrane (PM)-associated, light-activated protein kinases and, indeed, BL-induced autophosphorylation turned out to be a primary and essential step for the asymmetric growth response[5]. In this context, members of the 14-3-3 family were identified as phot1 interactors in *Arabidopsis*. Eukaryotic 14-3-3 proteins are known to interact with a multitude of polypeptides in a phosphorylation-dependent manner, thereby regulating distinct cellular processes[6]. Plant 14-3-3s are crucial components regulating auxin transport-related development and polarity of PIN-FORMED (PIN) auxin efflux carriers[7]. As yet, however, a functional role of phot1/14-3-3 association could not be proven[5,8]. Furthermore, evidence for *trans*-phosphorylation activity of phototropins is surprisingly limited. Besides BLUE LIGHT SIGNALING 1[9] and CONVERGENCE OF BLUE LIGHT AND CO$_2$ 1[10], both of which contribute to regulation of BL-induced stomatal opening, ATP-BINDING CASSETTE B19[11] and PHYTOCHROME KINASE SUBSTRATE 4[12,13] have been shown to be direct substrate targets of phot1. The last two are indirectly or directly involved in regulating phototropism.

The polar localization of PIN proteins within the PM made them likely candidates promoting formation of the auxin gradient that precedes phototropic growth[14]. Indeed, a mutant lacking the three major PINs expressed in aerial plant parts (PIN3, PIN4, and PIN7) is severely compromised in phototropism[15]. Unilateral illumination polarizes PIN3 specifically to the inner lateral side of hypocotyl endodermis cells, aligning PIN3 polarity with the light direction and presumably redirecting auxin flow towards the shaded side[16]. Moreover, the activity of PINs is positively regulated by two protein kinase families from the AGCVIII class, namely PINOID and D6 PROTEIN KINASES[17]. Although phototropins belong to the same kinase class, direct PIN phosphorylation could not be demonstrated[16]. Taken together, signaling events that couple photoreceptor activation to changes in PIN polarization and consequently auxin relocation remain mainly elusive.

In this regard, the PM-associated NON-PHOTOTROPIC HYPOCOTYL 3 (NPH3) might represent a promising component of early phototropic signaling events. It acts downstream of the photoreceptors and appears to be instrumental for auxin redistribution[3,4,18,19]. NPH3 possesses—in addition to the central NPH3 domain—two putative protein–protein interaction domains, a C-terminal coiled-coil (CC) domain, and an N-terminal bric-a-brac, tramtrack and broad complex (BTB) domain[1,20] (Supplementary Fig. 1). Indeed, NPH3 physically interacts not only with the photoreceptor phot1 but also with further early signaling elements, such as ROOT PHOTOTROPISM 2 (RPT2)[21]—another member of the plant-specific NPH3/RPT2-like (NRL) family—and defined members of the PHYTOCHROME KINASE SUBSTRATE (PKS) family[22,23]. Interestingly, NPH3 exists in a phosphorylated form in dark-grown seedlings and becomes rapidly dephosphorylated upon phot1 activation[24,25]. Later on, the alteration in phosphorylation status was shown to correlate closely with light-driven changes in the subcellular localization of NPH3, which detaches from the PM upon irradiation, forming aggregated particles in the cytosol[26]. As found for light-triggered dephosphorylation[24], formation of the NPH3 particles is reversible upon darkness or prolonged irradiation[26]. One factor required for the recovery of phosphorylated NPH3 at the PM over periods of prolonged irradiation is its interaction partner RPT2[26]. Altogether, this has led to the current model that the phosphorylation status of NPH3 determines its subcellular localization and function: phosphorylation of NPH3 promotes its action in mediating phototropic signaling from the PM, whereas NPH3 dephosphorylation reduces it by internalizing NPH3 into aggregates[4,18,26,27]. As yet, however, the functional significance of NPH3 (de)phosphorylation remains poorly understood[25,28].

Here we identified members of the 14-3-3 family as novel interactors and major regulators of NPH3. Our analyses revealed that BL induces phosphorylation of the third last NPH3 residue (S744), which in turn enables 14-3-3 association. Complex formation interferes with the ability of NPH3 to bind to polyacidic phospholipids, resulting in its displacement from the PM. Accumulation of NPH3 in the cytosol causes formation of membraneless condensates. Intriguingly, both PM association and 14-3-3-triggered PM dissociation are required for NPH3 function. Taking the reversibility of the light-induced processes into account, the phototropin-triggered and 14-3-3-mediated dynamic change in the subcellular localization of NPH3 seems to be crucial for its proper function in the phototropic response.

## Results

**PM association of NPH3 is phospholipid-dependent and requires its C-terminal domain.** Although NPH3 is hydrophilic in nature, green fluorescent protein (GFP) tagged NPH3 (GFP:NPH3) (35S or native promoter) localized to the cell periphery in the leaf epidermis of transiently transformed and dark-adapted *Nicotiana benthamiana* (Fig. 1a and Supplementary Fig. 2c), suggesting PM association as described previously[1,26,27]. As yet, the molecular mechanism of NPH3 membrane recruitment in darkness remains elusive. MACCHI-BOU 4 (MAB4)/ENHANCER OF PINOID (ENP), another member of the NRL family, was recently shown to associate with the PM in a PIN-dependent manner[29]. Besides protein–protein interactions, hydrophobic and protein–lipid interactions can cause membrane anchoring of proteins. Several members of the AGCVIII kinase class—although not phot1—contain a basic and hydrophobic (BH) motif in the middle domain of the kinase. This polybasic motif interacts directly with phospholipids and is required for PM binding[30]. When we applied the BH score prediction[31] to NPH3, two putative BH motifs were identified in its C-terminal domain (Supplementary Fig. 2a). To examine the importance of electronegativity for NPH3 PM association in the dark, we made use of a genetic system that depletes the polyacidic phosphoinositide (PI) phosphatidylinositol-4-phosphate (PI4P) at the PM via lipid anchoring (myristoylation and palmitoylation (MAP)) of the catalytic domain of the yeast SAC1 PI4P phosphatase[32,33]. Transient co-expression of GFP:NPH3 together with MAP:mCherry:SAC1, but not the catalytically inactive version MAP:mCherry:SAC1$_{DEAD}$, displaced NPH3 from the PM into discrete cytosolic bodies in darkness (Fig. 1a), reminiscent of the aggregated particles that have been observed upon BL treatment[26,27]. The strong and unique electrostatic signature of the plant PM is powered by the additive effect of PI4P and the phospholipids phosphatidic acid (PA) and phosphatidylserine (PS)[33–36]. In lipid overlay assays, hemagglutinin-tagged NPH3 (HA:NPH3) bound to several phospholipids characterized by polyacidic headgroups, namely PA as well as the PIs PI3P, PI4P, PI5P, PI(3,4)P$_2$, PI(3,5)P$_2$, PI(4,5)P$_2$, and PI(3,4,5)P$_3$ (Fig. 1b). HA:NPH3 did neither bind to phospholipids with monoacidic headgroups, such as phosphatidylinositol or PS, nor to phospholipids with neutral headgroups, namely phosphatidylcholine (PC) and phosphatidylethanolamine (PE). Deletion of the C-terminal 51 residues of NPH3 (HA:NPH3ΔC51, still comprising the CC

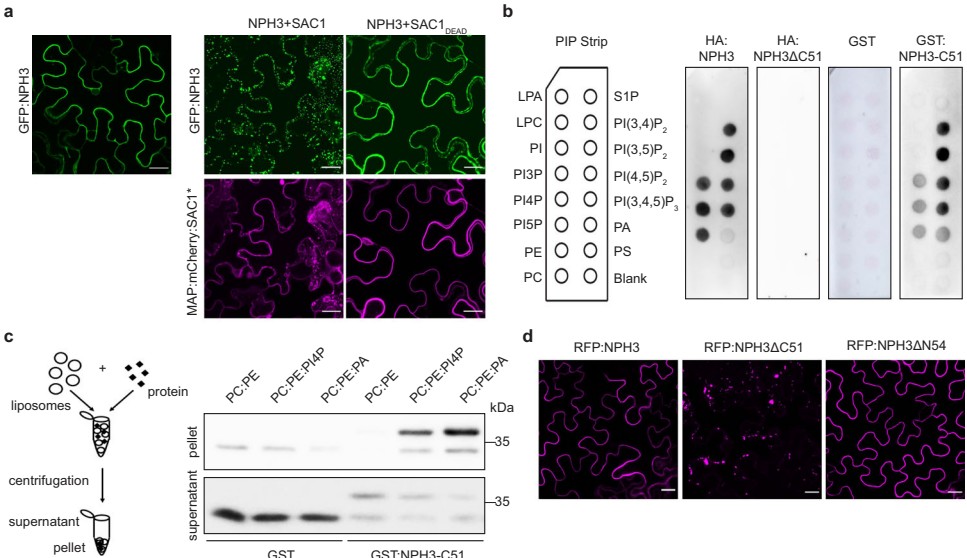

**Fig. 1 NPH3 binds to polyacidic phospholipids via its C-terminal domain. a** Representative confocal microscopy images of leaf epidermal cells from dark-adapted *N. benthamiana* transiently co-expressing GFP:NPH3 and either MAP:mCherry:SAC1 or MAP:mCherry:SAC1$_{DEAD}$. Expression was driven by the 35S promoter. *Z*-stack projection of GFP:NPH3 co-expressed with MAP:mCherry:SAC1 is shown. Single expression of GFP:NPH3 is shown as control. Scale bars, 25 μm. **b** Lipid overlay assay performed with either in vitro-transcribed and -translated HA:NPH3 and HA:NPH3ΔC51 or purified GST and GST:NPH3-C51. Immunodetection was conducted by using anti-HA or anti-GST antibodies, respectively. See main text for abbreviations of lipids. **c** Liposome-binding assay performed with purified GST or GST:NPH3-C51 and large unilamellar liposomes containing the neutral phospholipids PE and PC mixed with either the polyacidic PI4P or PA as specified. Anti-GST immunoblot of pellet and supernatant is shown (lower band corresponds to GST, upper band to GST:NPH3-C51). **d** Representative confocal microscopy images of leaf epidermal cells from dark-adapted *N. benthamiana* transiently expressing RFP:NPH3ΔC51 (*Z*-stack projection) or RFP:NPH3ΔN54. Expression was driven by the 35S promoter. RFP:NPH3 is shown as control. Scale bars, 25 μm. All experiments were performed at least three times with similar results.

domain, Supplementary Fig. 1) abolished lipid binding, whereas the bacterially expressed C-terminal 51 residues of NPH3 (tagged with glutathione S-transferase (GST), GST:NPH3-C51) turned out to be sufficient to bind to polyacidic phospholipids (Fig. 1b). Moreover, GST:NPH3-C51 bound to large unilamellar liposomes containing the polyacidic phospholipids PI4P or PA, but not to liposomes composed of only neutral phospholipids such as PC and PE (Fig. 1c). Apparently, the C-terminal 51 residues of NPH3 enable electrostatic association with membrane bilayers irrespective of posttranslational protein modifications or association with other proteins. As expected, transient expression of red fluorescent protein (RFP) or GFP-tagged NPH3ΔC51 in *N. benthamiana* (35S or native promoter) revealed loss of PM recruitment in the dark, as evident by the presence of discrete bodies in the cytosol (Fig. 1d and Supplementary Fig. 2c). This resembles the scenario observed upon co-expression of NPH3 and SAC1 (Fig. 1a), as well as upon transient expression of NPH3ΔC65:GFP in guard cells of *Vicia faba*[37]. By contrast, deletion of the N-terminal domain (*35S::RFP:NPH3ΔN54* or *NPH3::GFP:NPH3ΔN54*, still comprising the BTB domain, Supplementary Fig. 1) did not affect PM association of NPH3 in darkness (Fig. 1d and Supplementary Fig. 2c).

**An amphipathic helix is essential for phospholipid binding and PM association of NPH3 in vivo.** As already mentioned, two polybasic motifs with a BH score above the critical threshold value of 0.6 (window size 11 as recommended for the detection of motifs closer to the termini[31]) were identified in the C-terminal domain of NPH3: (i) a R-rich motif (R736-R742) close to the C-terminal tail and (ii) a K-rich motif further upstream (W700-M713) (Fig. 2a and Supplementary Fig. 2a). The latter is predicted to form an amphipathic helix, organized with clearly distinct positively charged and hydrophobic faces. The hydrophobic moment—a measure of the amphiphilicity—was calculated to be 0.58 (Supplementary Fig. 2b),

similar to the PM anchor of Remorin[38]. In order to test the requirement of the two motifs for membrane association, NPH3 mutant variants were generated in both HA:NPH3 and GST:NPH3-C51. Within the R-rich motif, all five basic amino acids were replaced by alanine (NPH3-5KR/A). Furthermore, both hydrophobicity and positive charge of the amphipathic helix were decreased by exchange of four hydrophobic residues (NPH3-4WLM/A) and of four lysine residues (NPH3-4K/A), respectively (Fig. 2a and Supplementary Fig. 2b). The ability of any of the three NPH3 replacement variants to bind polyacidic phospholipids in vitro was significantly impaired (Fig. 2b, c). Nonetheless, the RFP:NPH3-5KR/A mutant remained PM-associated in the dark when transiently expressed in *N. benthamiana* (Fig. 2d). To verify that the terminal R-rich motif is dispensable for PM recruitment in vivo, NPH3 was truncated by the C-terminal 28 residues (*35S::RFP:NPH3ΔC28* or *NPH3::GFP:NPH3ΔC28*). Indeed, PM anchoring was unaffected (Fig. 2d and Supplementary Fig. 2d). By contrast, modification of either the amphiphilicity (*35S::RFP:NPH3-4K/A* or *NPH3::GFP:NPH3-4K/A*) or the hydrophobicity (*35S::RFP:NPH3-4WLM/A* or *NPH3::GFP:NPH3-4WLM/A*) of the amphipathic helix gave rise to cytosolic particle-like structures in darkness (Fig. 2d and Supplementary Fig. 2d). Although these particles differ in shape and size, strict colocalization of the respective NPH3 variants was observed upon co-expression (Supplementary Fig. 3). Taken together, these experiments revealed the necessity of the amphipathic helix for PM anchoring in vivo and indicate hydrophobic interactions to also contribute to PM association of NPH3. Thus, one attractive hypothesis is that the positively charged residues interact electrostatically with polyacidic phospholipids of the PM followed by partial membrane penetration. By this means, interactions with both the polar headgroups and the hydrocarbon region of the bilayer would be established in darkness, causing anchor properties of NPH3 similar to intrinsic proteins.

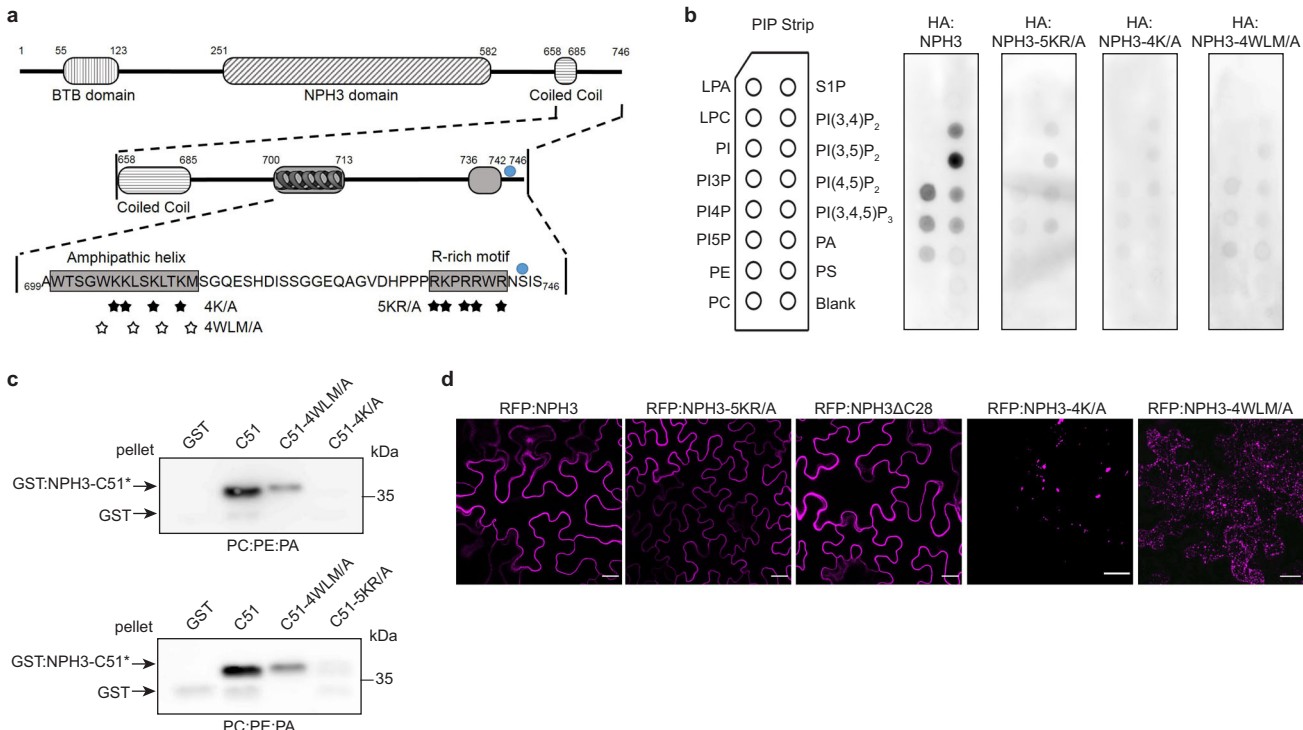

**Fig. 2 An amphipathic helix within the C-terminal domain is required for NPH3 phospholipid binding, membrane association, and plasma membrane localization. a** Domain structure and primary sequence of NPH3 showing the two putative BH domains (amphipathic helix and R-rich motif) within the C-terminal region. Stars depict residues substituted by alanine (A) in the NPH3 variants, blue circle depicts the 14-3-3-binding site (see Fig. 3). **b** Lipid overlay assay performed with in vitro-transcribed and -translated HA:NPH3 and HA:NPH3 variants characterized by the substitutions depicted in **a**. Immunodetection was conducted by using the anti-HA antibody. **c** Liposome-binding assay performed with purified GST and GST:NPH3-C51 variants (referred to as C51) characterized by the substitutions depicted in **a**. Large unilamellar liposomes containing the neutral phospholipids PE and PC mixed with the polyacidic PA were used. Anti-GST immunoblot of the pellet (see Fig. 1c) is shown. **d** Representative confocal microscopy images of leaf epidermal cells from dark-adapted *N. benthamiana* transiently expressing RFP:NPH3 variants characterized by the substitutions depicted in **a** as well as RFP:NPH3ΔC28. Expression was driven by the 35S promoter. Z-stack projections of RFP:NPH3-4K/A and RFP:NPH3-4WLM/A are shown. RFP:NPH3 is shown as control. Scale bars, 25 μm. Experiments **b**–**d** were performed at least three times with similar results.

**14-3-3 Proteins interact with NPH3 via a C-terminal binding motif in a BL-dependent manner.** A yeast two-hybrid screen performed in our lab (see ref. [39]) identified NPH3 as a putative interactor of several *Arabidopsis* 14-3-3 isoforms, among those representatives of both phylogenetic 14-3-3 groups, the non-epsilon group (isoform omega, Fig. 3a) and the epsilon group (isoform epsilon, Supplementary Fig. 4a)[7]. In contrast to phot1[8], 14-3-3 isoform specificity was thus not observed for binding to NPH3. Complex formation of NPH3 and 14-3-3 omega was confirmed in planta by co-immunoprecipitation (CoIP) of fluorophore-tagged proteins transiently co-expressed in *N. benthamiana* leaves (Fig. 3b). To elucidate the impact of light on 14-3-3/NPH3 complex assembly, transgenic *Arabidopsis* lines expressing 14-3-3 epsilon:GFP under control of the native promoter[7] and, as control, UBQ10::GFP were employed. Three-day-old etiolated seedlings were either maintained in complete darkness or irradiated with BL (1 μmol m$^{-2}$ s$^{-1}$) for 30 min. Potential targets of 14-3-3 epsilon:GFP were identified by stringent CoIP experiments coupled with mass spectrometry (MS)-based protein identification. As expected, several known 14-3-3 clients[7] were detected by MS and, remarkably, NPH3 emerged as a major 14-3-3 interactor (Supplementary Table 1). Binding capability of characterized 14-3-3 targets, such as the H$^+$-ATPase (AHA1 and AHA2) and cytosolic invertase 1, was not modified by BL treatment. By contrast, NPH3 turned out to be a BL-dependent 14-3-3 interactor in planta (Fig. 3c and Supplementary Table 1). CoIP of fluorophore-tagged proteins transiently co-

expressed in *N. benthamiana* leaves confirmed that physical association of NPH3 and 14-3-3 omega is not detectable in darkness, whereas BL irradiation triggers complex formation (Fig. 3d). Assuming 14-3-3 association to depend on phosphorylation of the target protein, this observation is in apparent contrast to the light-induced dephosphorylation of NPH3[24].

The specific phosphorylatable 14-3-3-binding sequences of numerous target proteins are mostly flexible and disordered[40]. As both the N- and C-terminal domain of NPH3 are predicted to be intrinsically disordered (Supplementary Fig. 1[41]), the corresponding truncated versions were analyzed by yeast two-hybrid assays. While NPH3ΔN54 was capable of 14-3-3 binding, deletion of the C-terminal 51 residues (NPH3ΔC51) abolished 14-3-3 association, suggesting that the 14-3-3-binding site—in addition to the membrane targeting motif—localizes downstream of the CC domain (Fig. 3a and Supplementary Fig. 4a). We therefore exchanged amino acid residues, phosphorylation of which has recently been demonstrated in planta (S722, S723, S744, and S746[42,43]), for a non-phosphorylatable alanine. Strikingly, 14-3-3 binding was not affected in all but one NPH3 mutant: replacement of S744—the third last residue of NPH3—prevented 14-3-3 association both in yeast (Fig. 3a and Supplementary Fig. 4a) and in planta (Fig. 3b), suggesting a phosphorylation-dependent C-terminal 14-3-3-binding motif (pS/pTX$_{1-2}$-COOH)[44] in NPH3. Phosphomimic variants (NPH3-S744D/S744E), however, do not allow for 14-3-3 binding (Fig. 3a and Supplementary Fig. 4a), consistent with the general finding that

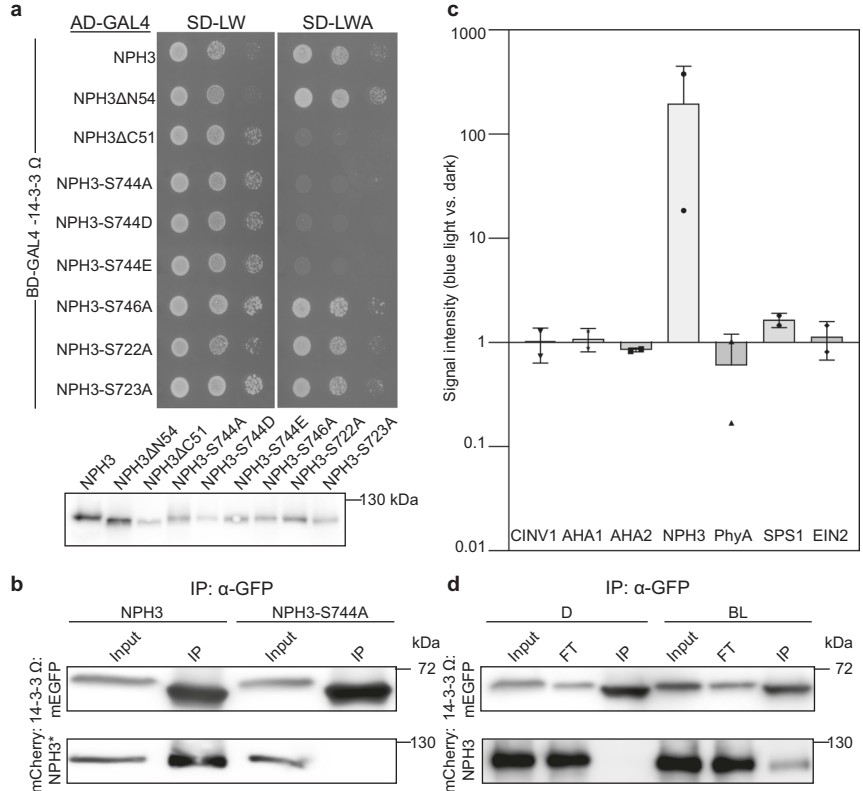

**Fig. 3 Interaction of NPH3 and 14-3-3 proteins is triggered by blue light irradiation and abolished by mutation of the third last NPH3 residue. a** Yeast two-hybrid interaction analysis of the *Arabidopsis* 14-3-3 isoform omega with NPH3 variants (upper panel). Expression of the diverse NPH3 fusion proteins was confirmed by anti-HA immunoblot of yeast extracts (lower panel). AD, activating domain, BD, binding domain. **b** In vivo interaction of 14-3-3 omega:mEGFP and either mCherry:NPH3 or mCherry:NPH3-S744A in transiently transformed *N. benthamiana* leaves. Expression was driven by the 35S promoter. Freshly transformed tobacco plants were kept under constant light for 42 h. The crude extract was immunoprecipitated using GFP beads. Input and immunoprecipitate (IP) were separated on 11% SDS-PAGE gels, followed by immunoblotting with anti-GFP and anti-RFP antibodies, respectively. **c** *Arabidopsis* 14-3-3 epsilon interactors were identified by mass spectrometry analysis of anti-GFP immunoprecipitations from etiolated seedlings expressing 14-3-3 epsilon:GFP and either maintained in darkness or irradiated with blue light (BL) (1 µmol m$^{-2}$ s$^{-1}$) for 30 min (two biological replicates). Expression was driven by the native promoter. Protein intensities of 14-3-3 client proteins were normalized to relative abundance of the bait protein (Supplementary Table 1). Fold changes in relative abundance (mean ± SD, logarithmic scale) of blue light treatment vs. darkness are given. AHA1, AHA2, *Arabidopsis* H$^+$-ATPase; CINV1, cytosolic invertase 1; EIN2, ethylene insensitive 2; PhyA, phytochrome A; SPS1, sucrose phosphate synthase 1. **d** In vivo interaction of 14-3-3 omega:mEGFP and mCherry:NPH3 in transiently transformed *N. benthamiana* leaves. Expression was driven by the 35S promoter. Dark-adapted tobacco plants were either kept in darkness (D) or treated with BL (10 µmol m$^{-2}$ s$^{-1}$) for 40 min. The crude extract was immunoprecipitated using GFP beads. Input, flowthrough (FT) and IP were separated on 11% SDS-PAGE gels, followed by immunoblotting with anti-GFP and anti-RFP antibodies, respectively. Experiments in **a**, **b**, and **d** were performed at least three times with similar results.

aspartate and glutamate do not provide good phosphomimetic residues with respect to 14-3-3 binding[45]. Considering that constitutive 14-3-3 complex formation of other plant targets characterized by a C-terminal binding site, such as the H$^+$-ATPase or the transcription factor FD, has been observed in yeast[46,47], light-independent NPH3/14-3-3 interaction in yeast (Fig. 3a and Supplementary Fig. 4a) might arise from a promiscuous kinase with a certain preference for terminal motifs.

**14-3-3 Association is required for NPH3 function and its BL-induced PM dissociation.** To address the functional significance of 14-3-3 association in vivo, GFP-tagged NPH3 variants were expressed in a T-DNA-induced loss-of-function allele of *NPH3*, *nph3-7*[48]. GFP:NPH3 was fully functional in restoring the severe impairment of hypocotyl phototropism in *nph3-7*, regardless of whether expression was driven by the native or the 35S promoter (Fig. 4a and Supplementary Fig. 4b), thus confirming previous data[26,27]. By contrast, phototropic hypocotyl bending was still significantly reduced when NPH3 incapable of 14-3-3 association (GFP:NPH3-S744A) was expressed (Fig. 4a and Supplementary

Fig. 4b), suggesting that BL-induced interaction with 14-3-3 is required for proper NPH3 function.

Both GFP:NPH3 and GFP:NPH3-S744A localized to the cell periphery in the hypocotyl of etiolated transgenic seedlings (Fig. 4b and Supplementary Fig. 4c). Within minutes, however, the BL laser used to excite GFP (488 nm, activates phototropins), induced detachment of GFP:NPH3 from the PM into discrete bodies/particle-like structures in the cytoplasm (Supplementary Movie 1). This BL-induced shift in subcellular localization is mediated by phot1 activity[26] and, again, could be observed independent of whether expression of GFP:NPH3 was under control of the endogenous (Supplementary Fig. 4c[27]) or the 35S promoter (Fig. 4b[26]). By contrast, GFP:NPH3-S744A remained mainly PM-associated upon irradiation (Fig. 4b, Supplementary Fig. 4c, Supplementary Fig. 4e, and Supplementary Movie 2). Mutation of the 14-3-3-binding site does thus not affect PM association of NPH3 in darkness but prevents BL-triggered PM dissociation, suggesting that light-induced binding of 14-3-3 proteins to the third last, presumably phosphorylated residue S744 is required to internalize NPH3 from the PM into cytosolic

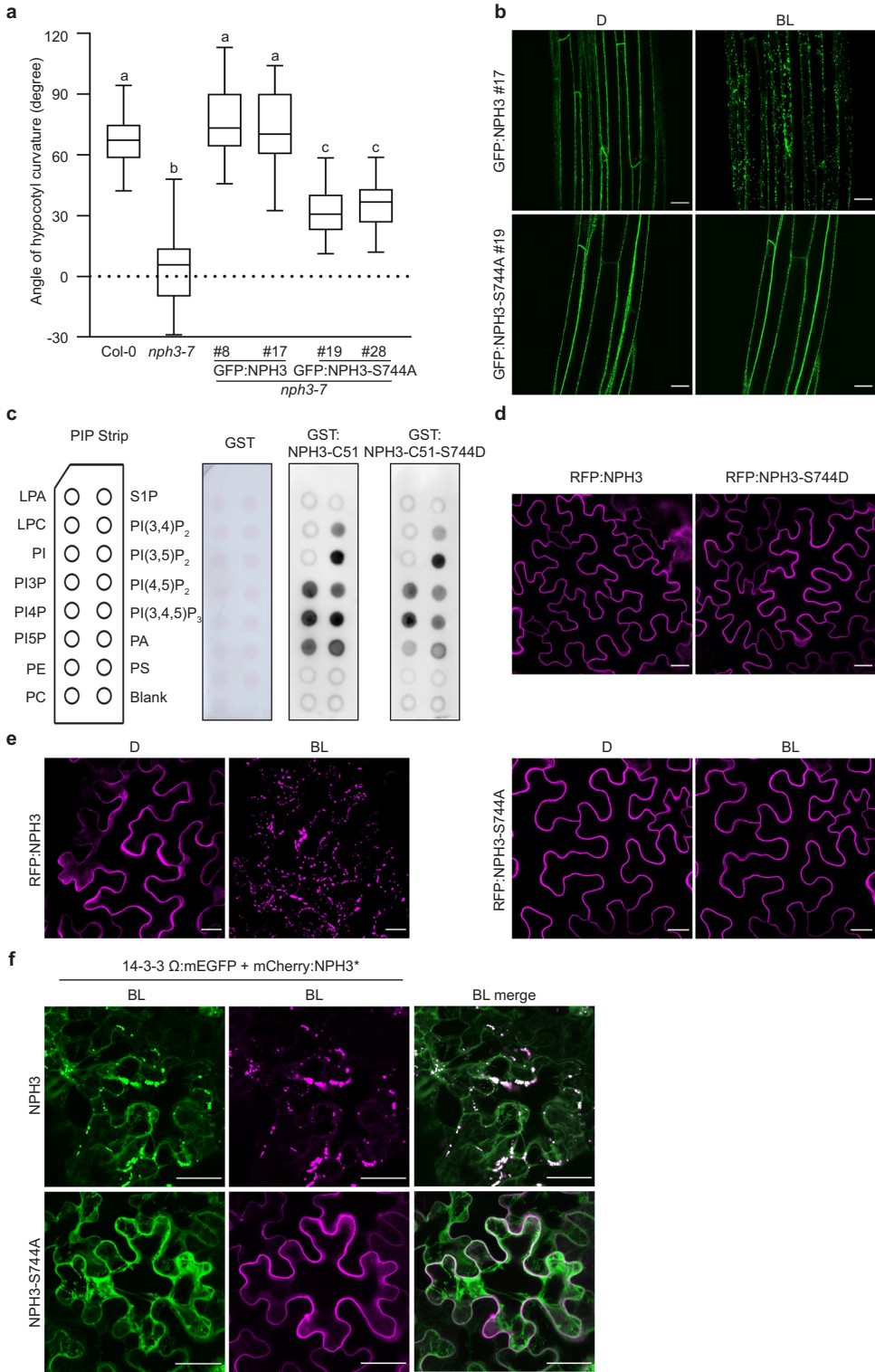

particles. Nonetheless, the suspected phosphorylation of S744 might per se decrease the interaction of NPH3 with polyacidic phospholipids, hence triggering the PM dissociation. Yet, the appropriate phosphomimic version of NPH3 (S744D) was neither impaired in phospholipid interaction in vitro (GST:NPH3-C51-S744D, Fig. 4c) nor PM recruitment in vivo (RFP:NPH3-S744D, Fig. 4d). Altogether, the C-terminal domain serves a dual function in determining the subcellular localization of NPH3, as it comprises both the amphipathic helix required for phospholipid-dependent PM association in

darkness and the 14-3-3-binding motif mediating BL-triggered PM dissociation.

We confirmed our findings in transiently transformed *N. benthamiana* leaves (Fig. 4e, Supplementary Fig. 4d, and Supplementary Movies 3 and 4). Here, primarily RFP-tagged proteins were employed, as excitation of RFP (558 nm)—unlike GFP (488 nm)—does not activate phototropins. This enabled us to conditionally activate phot1 by means of the GFP laser (488 nm). It became evident that RFP:NPH3—instead of being directly internalized into discrete bodies—initially detaches from the PM

**Fig. 4 14-3-3 Binding is required for proper NPH3 function in phototropic hypocotyl bending and its light-triggered detachment from the plasma membrane. a** Quantification of hypocotyl phototropism (mean ± SD) in etiolated *Arabidopsis nph3-7* seedlings expressing either GFP:NPH3 or GFP:NPH3-S744A. Expression was driven by the 35S promoter. Seedlings were exposed for 24 h to unilateral blue light (BL) (1 µmol m$^{-2}$ s$^{-1}$) ($n \geq 30$ seedlings per experiment, one representative experiment of three replicates is presented). One-way ANOVA with Tukey's post hoc test is shown, different letters mark statistically significant differences ($P < 0.05$), same letters mark statistically nonsignificant differences. Center line: median, bounds of box: minima and maxima (25th and the 75th percentiles), whiskers: 1.5 × IQR (IQR: the interquartile range between the 25th and the 75th percentile). Exact *P*-values for all experiments are provided in the source data file. **b** Representative confocal microscopy images of hypocotyl cells from 3-day-old etiolated transgenic *Arabidopsis nph3-7* seedlings shown in **a**. Seedlings were either kept in darkness (D) or treated with BL (~6 min GFP laser). Scale bars, 25 µm. **c** Lipid overlay assay performed with purified GST, GST:NPH3-C51, and GST:NPH3-C51-S744D. Immunodetection was conducted by using the anti-GST antibody. **d** Representative confocal microscopy images of leaf epidermal cells from dark-adapted *N. benthamiana* transiently expressing RFP:NPH3-S744D. Expression was driven by the 35S promoter. RFP:NPH3 is shown as control. Scale bars, 25 µm. **e** Representative confocal microscopy images of leaf epidermal cells from *N. benthamiana* transiently expressing RFP:NPH3 or RFP:NPH3-S744A. Expression was driven by the 35S promoter. Dark-adapted tobacco plants were either kept in D or treated with BL (~11 min GFP laser). Z-stack projection of BL-treated RFP:NPH3 is shown. Scale bars, 25 µm. **f** Representative confocal microscopy images of leaf epidermal cells from *N. benthamiana* transiently co-expressing 14-3-3 omega:mEGFP and either mCherry:NPH3 or mCherry:NPH3-S744A. Expression was driven by the 35S promoter. Dark-adapted tobacco plants were treated with blue light BL (10 µmol m$^{-2}$ s$^{-1}$) for 40 min. Z-stack projections are shown. Scale bars, 25 µm. All experiments were performed at least three times with similar results.

and moves along cytoplasmic strands comparable to soluble polypeptides (Supplementary Movie 3). Body formation in the cytosol is initiated after a lag time of ~4–5 min. Generation of particle-like structures might thus depend on soluble NPH3, exceeding a critical concentration in the cytosol. Upon co-expression of mEGFP-tagged 14-3-3 omega, colocalization with mCherry:NPH3 was observed in such particles (Fig. 4f).

**NPH3 forms membraneless condensates in the cytosol**. BL-induced PM dissociation and particle assembly of RFP:NPH3 in the cytosol seem to be separate and consecutive processes (Supplementary Movie 3). As yet, the identity of these particles has not been determined. RFP:NPH3ΔC51 is devoid of the amphipathic helix and localized to cytosolic particles in darkness (Fig. 1d). Subcellular fractionation clearly illustrated that the lack of the C-terminal region shifts NPH3 from a membrane-associated state to the soluble fraction (Fig. 5a). This reveals a non-membrane-attached state of NPH3 in discrete bodies as has been suggested for NPH3 aggregates generated upon BL irradiation[26]. Apparently, the mechanisms of NPH3 targeting towards and away from the PM are distinct from vesicle-mediated transport of transmembrane proteins. This is in line with the observation that NPH3 is insensitive to an inhibitor of endosomal trafficking[26]. Considering the lack of the 14-3-3-binding motif in NPH3ΔC51, 14-3-3 association seems dispensable for NPH3 body formation in the cytosol. To confirm this assumption, we examined NPH3 variants incapable of 14-3-3 binding, namely (i) RFP:NPH3-4K/A-S744A and (ii) GFP:NPH3-S744A, the latter upon co-expression with MAP:mCherry:SAC1. Indeed, prevention of 14-3-3 association did not affect assembly of RFP:NPH3-4K/A-S744A particles in darkness (Fig. 5b, RFP:NPH3-4K/A is shown in Fig. 2d). Similar to GFP:NPH3 (Fig. 1a), GFP:NPH3-S744A localized to cytosolic particles in the dark upon co-expression of SAC1 but not SAC1$_{DEAD}$ (Fig. 5c). Generation of NPH3 particles is hence feasible in the absence of 14-3-3s and might be due to intrinsic properties of NPH3 when exceeding a critical concentration in the cytosol. Taking constitutive PM association of NPH3-S744A in the absence of SAC1 into account, 14-3-3 association seems to be crucial for initial PM detachment, while formation of discrete bodies in the cytosol occurs as an autonomous process.

The dynamic generation and morphology of NPH3 bodies (Supplementary Movie 3) is reminiscent of membraneless biomolecular condensates, which are micrometer-scale compartments in cells lacking surrounding membranes. An important organizing principle is liquid–liquid phase separation driven by multivalent macromolecular interactions—either mediated by

modular interaction domains or disordered regions[49]. NPH3 is characterized by both intrinsically disordered regions and interaction domains such as the BTB and the CC domain (Supplementary Fig. 1). We performed single-cell time-lapse imaging of RFP:NPH3 body formation to investigate whether NPH3 undergoes transition from a solute to a condensed state in *N. benthamiana*. Indeed, formation of particle-like structures in the cytosol is initiated after ~4 min and the fluorescence intensity per body gradually increased over time as a result of the growth in size (Fig. 5d, e). In contrast to the signal intensity, the number of bodies reached a maximum after ~10–15 min and afterwards started to decrease as a result of body fusion (Fig. 5d, f). Worth mentioning, these features are characteristic criteria of biomolecular condensates[49,50].

**Phosphorylation of the 14-3-3-binding site in NPH3 is light-dependent and reversible**. In dark-grown seedlings, NPH3 exists as a phosphorylated protein irrespective of phot1 activity[24]. Light-induced dephosphorylation of NPH3 is almost a dogma in the literature. It has been recognized as a slight shift in electrophoretic mobility of NPH3 upon SDS-polyacrylamide gel electrophoresis (PAGE)[24] and requires—in accordance with the light-induced formation of particle-like structures in the cytosol[26]—the photoreceptor phot1. In the following, (de)phosphorylation of NPH3, represented by a modification of its electrophoretic mobility, will be referred to as "general" (de)phosphorylation of NPH3. Nonetheless, the data presented so far suggest that light-triggered and presumably S744 phosphorylation-dependent 14-3-3 association contributes to NPH3 function—an obvious antagonism to the "dogma of dephosphorylation". A phosphosite-specific peptide antibody ($\alpha$-pS744) was therefore established (antigen: $_{734}$PPRKPRRWRN-S(P)-IS$_{746}$) and an antibody against the unmodified peptide ($\alpha$-NPH3) served as control. Examination of GFP:NPH3 in either *N. benthamiana* leaves or transgenic *Arabidopsis* lines revealed the typical enhanced electrophoretic mobility upon BL excitation (Fig. 6), indicative of a "general" dephosphorylation[24–26]. Intriguingly, the $\alpha$-pS744 antibody recognized GFP:NPH3, but not GFP:NPH3-S744A, exclusively upon BL irradiation (Fig. 6). BL thus triggers two different post-translational modifications of NPH3: (i) the phosphorylation of the 14-3-3-binding site (S744) and (ii) a "general" dephosphorylation. Yet, neither of the modifications could be observed for GFP:NPH3-S744A (Fig. 6a). To uncover light-induced 14-3-3 association at the molecular level, an immunoprecipitation of either GFP:NPH3 or GFP:NPH3-S744A was conducted and combined with 14-3-3 far-western analysis. Phosphorylation of S744 indeed enabled binding of purified recombinant 14-3-3 proteins to GFP:NPH3 but not GFP:NPH3-S744A upon SDS-PAGE (Fig. 6a, b). Prolonged

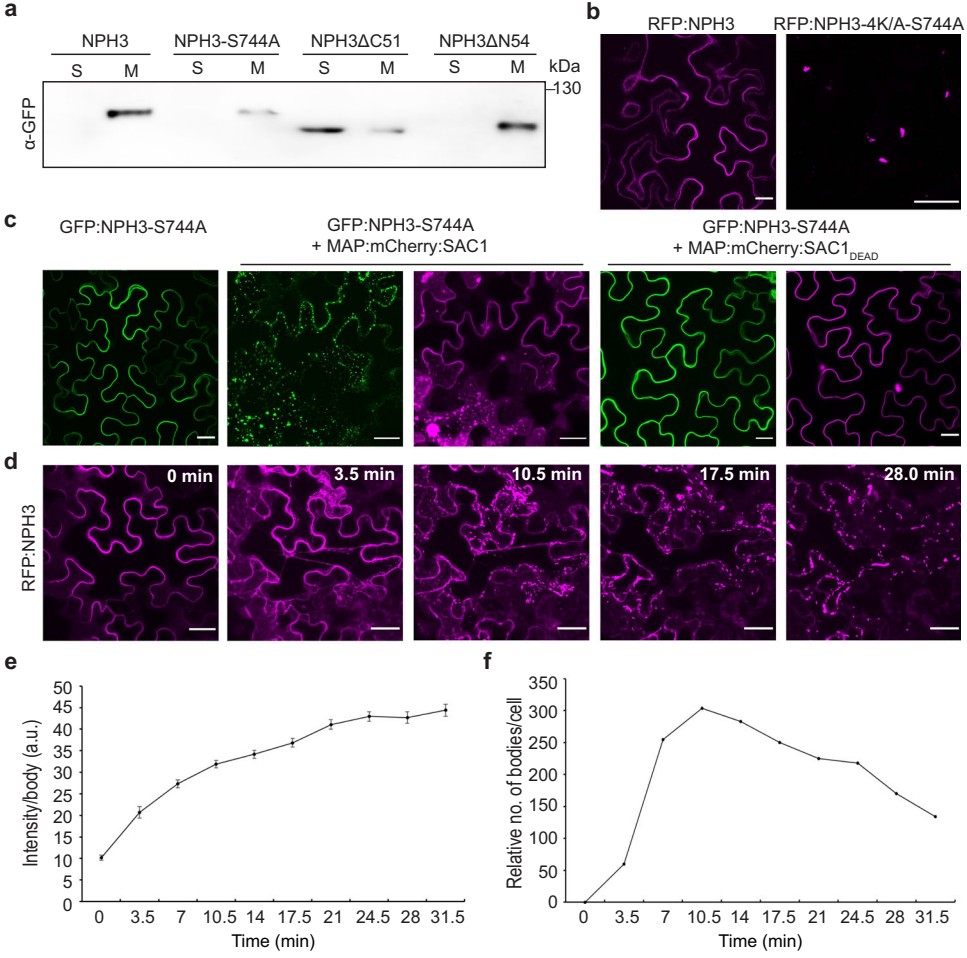

**Fig. 5 Formation of membraneless NPH3 condensates is independent of 14-3-3 binding. a** Representative anti-GFP immunoblots following subcellular fractionation of protein extracts prepared from dark-adapted *N. benthamiana* leaves transiently expressing GFP:NPH3 variants. Expression was driven by the 35S promoter. Proteins in each fraction (7.5 µg) were separated on 7.5% SDS-PAGE gels. It is noteworthy that the total amount of soluble proteins (S) is ~15 times higher as compared to the total amount of microsomal proteins (M) after 100,000 × g centrifugation. **b** Representative confocal microscopy images of leaf epidermal cells from dark-adapted *N. benthamiana* transiently expressing RFP:NPH3-4K/A-S744A. *Z*-stack projection is shown. Expression was driven by the 35S promoter. RFP:NPH3 is shown as control. Scale bars, 25 µm. **c** Representative confocal microscopy images of leaf epidermal cells from dark-adapted *N. benthamiana* transiently co-expressing GFP:NPH3-S744A and either MAP:mCherry:SAC1 or MAP:mCherry:SAC1$_{DEAD}$. Expression was driven by the 35S promoter. *Z*-stack projection of GFP:NPH3-S744A co-expressed with MAP:mCherry:SAC1 is shown. Single expression of GFP:NPH3-S744A is shown as control. Scale bars, 25 µm. **d–f** Single-cell time-lapse imaging of RFP:NPH3 condensation induced by GFP laser treatment of transiently transformed and dark-adapted *N. benthamiana*. Expression was driven by the 35S promoter. The image of time point 0 was taken in the absence of the GFP laser. *Z*-stack projections from selected time points of blue light treatment (**d**), fluorescence intensity per body (mean ± SEM) (**e**), and number of bodies (**f**) are shown. One representative experiment of five replicates is shown. Scale bars, 25 µm. Experiments **a–c** were performed at least three times with similar results.

irradiation or transfer of BL-irradiated seedlings to darkness is known to confer PM re-association of NPH3[26], correlating with a reduced electrophoretic mobility, indicative of a "general" re-phosphorylation[24,26]. Remarkably, we observed simultaneous dephosphorylation of S744 (Fig. 6b, c), effectively preventing binding of 14-3-3 to NPH3 (Fig. 6b). Taken together, the dark/light-dependent phosphorylation status of S744 determines 14-3-3 association with NPH3. In addition, the phosphorylation status of the 14-3-3-binding site and of NPH3 "in general" is modulated by the light regime in an opposite manner, giving rise to a coinciding, but inverse pattern. Time-course analyses, however, proved S744 phosphorylation of NPH3 to precede "general" dephosphorylation upon BL treatment (Fig. 6c). "General" dephosphorylation of NPH3 has been assumed to determine PM release of NPH3 coupled to particle assembly in the cytosol[4,18,26,27]. Our data now clearly indicate S744 phosphorylation-dependent 14-3-3 association to be

the cause of PM dissociation, but not of condensate assembly in the cytosol. "General" dephosphorylation might thus be coupled to PM dissociation and/or condensate formation. We examined the "general" phosphorylation status of both GFP:NPH3 and GFP:NPH3-S744A when co-expressed with SAC1. Despite the fact that both NPH3 variants constitutively localized to cytosolic condensates (Figs. 1a and 5c), GFP:NPH3 was phosphorylated in darkness and shifted to the dephosphorylated status upon BL treatment, whereas GFP:NPH3-S744A exhibited a permanent phosphorylated state (Fig. 6d). "General" dephosphorylation of NPH3 is thus not coupled to PM dissociation. Moreover, it is neither a prerequisite nor a consequence of condensate assembly, rather it requires prior light-triggered S744 phosphorylation and potentially 14-3-3 association (Fig. 6a, d). Taken together, we suggest that BL-induced phosphorylation of S744 provokes (i) 14-3-3 association, which releases NPH3 from the PM into the cytosol, and (ii) "general"

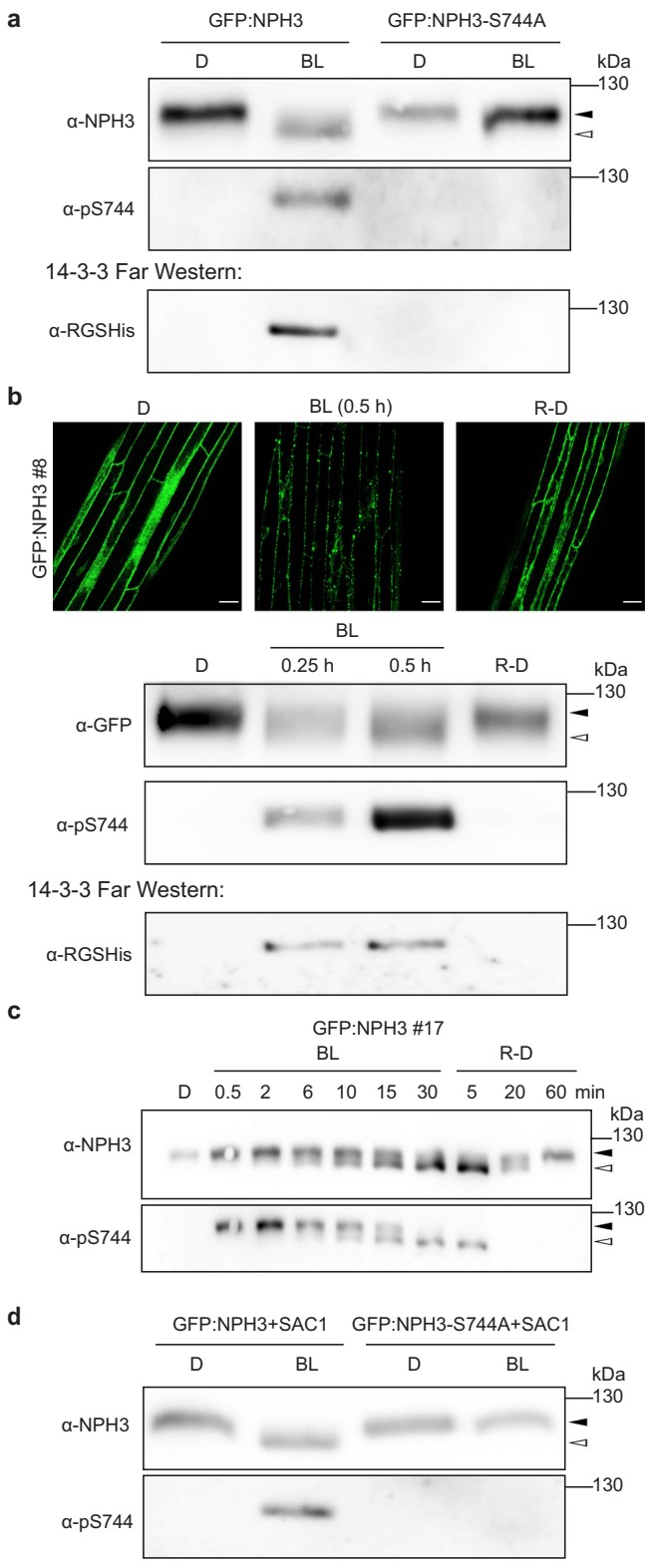

**Fig. 6 The phosphorylation status of the NPH3 14-3-3-binding site (S744) is dynamically modulated by the light regime.** The closed and open arrowheads indicate the positions of "generally" phosphorylated and dephosphorylated NPH3 proteins, respectively. **a** Immunoblot analysis and 14-3-3 far-western of anti-GFP immunoprecipitates from *N. benthamiana* leaves transiently expressing GFP:NPH3 or GFP:NPH3-S744A. Expression was driven by the 35S promoter. Dark-adapted tobacco plants were either maintained in darkness (D) or treated with blue light (BL) (10 μmol m$^{-2}$ s$^{-1}$) for 40 min. Immunoprecipitated proteins were separated on 7.5% SDS-PAGE gels. **b** Immunoblot analysis and 14-3-3 far-western of anti-GFP immunoprecipitates from *Arabidopsis nph3-7* expressing GFP:NPH3. Expression was driven by the 35S promoter. Three-day-old etiolated seedlings were treated with cycloheximide (100 μM) for 1 h and either maintained in D, treated with BL (1 μmol m$^{-2}$ s$^{-1}$) for the indicated time, or re-transferred to D (1 h) after 30 min of irradiation (R-D). Immunoprecipitated proteins were separated on 7.5% SDS-PAGE gels. The upper panel shows representative confocal microscopy images of hypocotyl cells from the seedlings under the specified conditions. Scale bars, 25 μm. **c** Immunoblot analysis of total protein extracts from *Arabidopsis nph3-7* ectopically expressing GFP:NPH3 (see **b**). Three-day-old etiolated seedlings were either maintained in D, treated with BL (1 μmol m$^{-2}$ s$^{-1}$) for the indicated time or re-transferred to D for the indicated time after 30 min of irradiation (R-D). Proteins were separated on 7.5% SDS-PAGE gels. **d** Immunoblot analysis of total protein extracts from *N. benthamiana* leaves transiently co-expressing MAP:mCherry:SAC1 and either GFP:NPH3 or GFP:NPH3-S744A. Expression was driven by the 35S promoter. Dark-adapted tobacco plants were either maintained in D or treated with BL (10 μmol m$^{-2}$ s$^{-1}$) for 40 min. Proteins were separated on 7.5% SDS-PAGE gels. All experiments were performed at least three times with similar results.

mediating phototropic signaling. In turn, NPH3 present in soluble condensates is considered to be inactive[18,26,27]. The functional relevance of the transient changes in subcellular NPH3 localization is, however, still not known. To assess the functionality of NPH3 variants constitutively localizing to condensates, GFP:NPH3-4K/A (RFP-tagged version shown in Fig. 2d), as well as GFP:NPH3ΔC51 (RFP-tagged version shown in Fig. 1d) were expressed in the loss-of-function *Arabidopsis* mutant *nph3-7*. Worth mentioning, the electrophoretic mobility of GFP:NPH3-4K/A corresponded to the dephosphorylated version of NPH3 and was not modified by light treatment (Fig. 7c), suggesting that "general" phosphorylation of NPH3 might take place at the PM. In line with the hypothesis mentioned above, NPH3 mutants constitutively present in condensates did not restore hypocotyl phototropism (Fig. 7a, b and Supplementary Movies 5 and 6). Contrary to the hypothesis, however, GFP:NPH3-S744A—despite exhibiting constitutive PM localization (Fig. 4b)—is also largely incapable of mediating phototropic hypocotyl bending in *nph3-7* (Fig. 4a). To verify significantly impaired activity of permanently PM-attached NPH3, we examined GFP:NPH3ΔC28 in addition. Comparable to the results obtained in *N. benthamiana* (RFP-tagged version shown in Fig. 2d), GFP:NPH3ΔC28 remained PM-associated upon activation of phot1 in stable transgenic *Arabidopsis* lines (Fig. 7b and Supplementary Movie 7) and its electrophoretic mobility was not modified by BL treatment (Fig. 7c). Noteworthy, both NPH3-S744A and NPH3ΔC28 still interacted with phot1 (Fig. 7d), indicating that complex formation at the PM is not compromised. Nevertheless, permanent attachment of NPH3 to the PM turned out to be insufficient for triggering the phototropic response in *nph3-7* (Fig. 7a).

Taken together, neither NPH3 mutants permanently detached from the PM nor NPH3 versions permanently attached to the PM seem to be fully functional (Fig. 7a, e). So what is the underlying mechanism of NPH3 function? We examined GFP:NPH3ΔN54 (RFP-tagged version shown in Fig. 1d and Supplementary Movie 8) in more detail. Similar to GFP:NPH3, GFP:NPH3ΔN54

dephosphorylation of NPH3. Formation of NPH3 condensates is, however, determined by the biological properties of PM-detached NPH3.

**Cycling of NPH3 might be key to function.** The light-triggered and reversible shift in subcellular localization of NPH3 has led to the hypothesis that PM localization of NPH3 promotes its action in

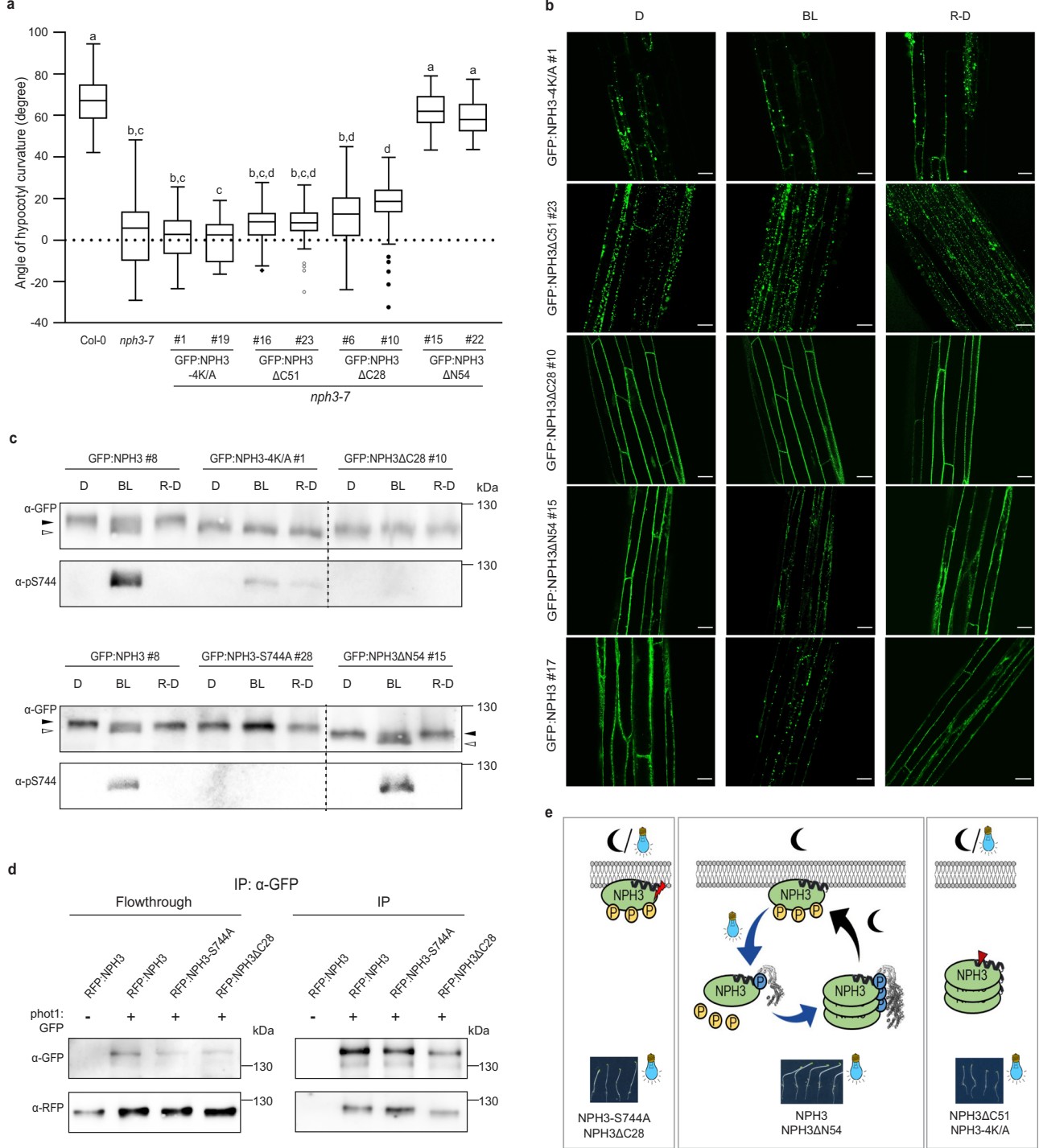

associated to the PM in etiolated seedlings (Fig. 7b). Upon irradiation, it (i) became phosphorylated at S744 (Fig. 7c), (ii) exhibited an increased electrophoretic mobility, indicative of a "general" dephosphorylation (Fig. 7c), and (iii) detached from the PM followed by condensate formation in the cytosol (Fig. 7b and Supplementary Movie 9). Furthermore, all these processes were reverted when seedlings were re-transferred to darkness (Fig. 7b, c). Intriguingly, expression of GFP:NPH3ΔN54 completely restored phototropic hypocotyl bending in nph3-7 (Fig. 7a) as did GFP:NPH3 (Fig. 4a). Thus, 14-3-3-mediated cycling of NPH3 between the PM and the cytosol might be of utmost importance for functionality (Fig. 7e).

## Discussion

Our data provide novel insight into the molecular mechanisms defining NPH3 function in BL-induced phototropic hypocotyl bending. We applied a combination of genetic, biochemical, physiological, and live-cell imaging approaches to uncover the impact of 14-3-3 proteins on NPH3, in particular its BL-triggered, S744 phosphorylation-dependent and functionally essential release from the PM. Association of NPH3 with the PM is known since decades, but how it is recruited to this compartment is unknown. We demonstrated that NPH3 attaches to the PM in a phospholipid-dependent manner in darkness (Fig. 1a). The electrostatic interaction with polyacidic phospholipids (Fig. 1b, c)

**Fig. 7 Functional relevance of the subcellular localization of NPH3. a** Quantification of hypocotyl phototropism (mean ± SD) in etiolated *Arabidopsis* nph3-7 seedlings expressing GFP:NPH3 variants (35S promoter) and exposed to unilateral blue light (BL) (1 μmol m$^{-2}$ s$^{-1}$, 24 h) (n ≥ 30 seedlings per experiment, three replicates). One-way ANOVA with Tukey's post hoc test is shown, different letters mark statistically significant differences (P < 0.05). Center line: median, bounds of box: minima and maxima (25th and the 75th percentiles), whiskers: 1.5 × IQR (IQR: the interquartile range between the 25th and the 75th percentile). Exact P-values for all experiments are provided in the source data file. **b** Representative confocal microscopy images of hypocotyl cells from seedlings shown in **a** and either maintained in darkness (D), treated with BL (~6 min GFP laser) or re-transferred to D (1 h) after 30 min BL (1 μmol m$^{-2}$ s$^{-1}$) (R-D). Scale bars, 25 μm. **c** Immunoblot analysis of total protein extracts (7.5% SDS-PAGE) from seedlings essentially treated as shown in **b**. BL treatment: 1 μmol m$^{-2}$ s$^{-1}$, 30 min. Dashed line: expected shift in molecular mass, closed/open arrowheads: positions of "generally" phosphorylated/dephosphorylated NPH3 proteins, respectively. **d** In vivo interaction of RFP:NPH3 variants and phot1:GFP in transiently transformed (35S promoter) and dark-adapted *N. benthamiana* leaves. Microsomal proteins were immunoprecipitated using GFP beads. Immunoblot analysis of flowthrough and immunoprecipitate (IP) (11% SDS-PAGE) is shown. All experiments were performed at least three times with similar results. **e** Model depicting the light regime-triggered changes in the phosphorylation status, subcellular localization, and phototropic responsiveness of NPH3. BL-induced and phosphorylation-dependent (S744, blue) 14-3-3 association releases NPH3 from the PM into the cytosol followed by condensate formation. Residues phosphorylated in darkness (yellow) and dephosphorylated upon BL cause an electrophoretic mobility shift ("general" (de)phosphorylation). Re-transfer to darkness reverts all BL-triggered processes, finally resulting in PM re-association (middle panel). NPH3 variants either constitutively attached to (red flash, left panel) or constitutively detached from (red arrowhead, right panel) the PM are non-functional. Cycling of NPH3 between the PM and the cytosol is suggested to be essential for proper function.

is mediated by four basic residues of an amphipathic helix, the hydrophobic face of which further contributes to PM association (Fig. 2d). We therefore suggest the amphipathic helix to be embedded in the PM inner leaflet with its hydrophobic interface inserted in the hydrophobic core of the bilayer, while the positively charged interface is arranged on the PM surface, interacting with the lipid polar heads. The molecular mechanism underlying PM association of NPH3 is thus different from the NRL protein MAB4/ENP, which is recruited to the PM by interaction with PIN proteins[29]. The amphipathic helix of NPH3 (amino acids 700–713) localizes downstream of the CC domain of NPH3 in its C-terminal region, which also encompasses the 14-3-3-binding site (S744) (Fig. 2a).

We discovered that BL induces two distinct posttranslational modifications in NPH3 (Fig. 6): (i) the immediate phosphorylation of S744, which in turn enables association of 14-3-3 proteins with NPH3, followed by (ii) the well-described dephosphorylation, represented by an enhanced electrophoretic mobility of NPH3 ("general" dephosphorylation)[24–26]. The—as yet unrecognized—BL-induced NPH3 phosphorylation event linked to 14-3-3 association is of utmost importance, as it is essential for (i) the BL-triggered internalization of NPH3 from the PM (Fig. 4b) and (ii) the function of NPH3 in phototropic hypocotyl bending (Fig. 4a). However, expression of NPH3-S744A, which is incapable of 14-3-3 interaction, partially restored the severe impairment of hypocotyl phototropism in nph3-7 (Fig. 4a). Residual functionality might be due to co-action of this constitutively PM-associated NPH3 mutant with certain members of the NRL protein family. Indeed, RPT2 is required for hypocotyl phototropism at light intensities utilized in our assays[26] and its expression is induced and stabilized by BL treatment[51]. The closest homolog of NPH3, DEFECTIVELY ORGANIZED TRIBUTARIES 3 (DOT3)[18] is, as yet, functionally uncharacterized. Worth mentioning, RPT2, DOT3, and also MAB4/ENP are capable of interacting with 14-3-3 isoforms representing the two phylogenetic 14-3-3 groups (Supplementary Fig. 5). In each case, exchange of the third last residue (serine) abolished 14-3-3 association in yeast (Supplementary Fig. 5), suggesting that phosphorylation-dependent 14-3-3 binding is not limited to NPH3 but rather represents a more widespread mechanism of NRL regulation. However, residual activity of NPH3-S744A in phototropic hypocotyl bending might alternatively be caused by its permanent association with the PM per se. Light treatment could induce a reorganization of NPH3-S744A within/along the PM, which might allow for phototropic responsiveness to a

certain level. Addressing these alternatives represents a formidable challenge for future research.

NPH3 has been described to re-localize directly from the PM into discrete bodies in the cytosol upon light treatment[26,27]. It became, however, evident that it initially detaches from the PM into the cytosol (Supplementary Movie 3). Here, NPH3 undergoes a dynamic transition from a dilute to a condensed state, resulting in the formation of membraneless biomolecular compartments (Fig. 5a, d). Biomolecular condensates are emerging as an important concept in signaling[52]. Their formation can be driven by multivalent interactions with other macromolecules, by intrinsically disordered regions within a single molecule, or both[49,53]. Interestingly, 14-3-3 proteins are dispensable for condensate assembly in the cytosol, as demonstrated by 14-3-3 binding-deficient NPH3 variants (Fig. 5b, c). Further studies will reveal whether condensate formation of the PM-detached NPH3 is essential for its action.

As described above, the light-triggered modifications of the phosphorylation pattern of NPH3 are highly complex. Our observations disproved the view that BL-triggered "general" dephosphorylation events determine PM dissociation of NPH3[18,26,27]. First of all, dephosphorylation of NPH3—i.e., a decrease in negative charge—is entirely inappropriate to interfere with membrane association relying on electrostatic interactions with polyacidic phospholipids. Furthermore, investigation of the seven NPH3 phosphorylation sites that were recently identified in etiolated *Arabidopsis* seedlings revealed that the phosphorylation status of these NPH3 residues was neither required for PM association in darkness nor BL-induced release of NPH3 into the cytosol[28]. By contrast, single-site mutation of the 14-3-3-binding site in NPH3 (S744A) abolished PM dissociation upon BL treatment (Fig. 4b, e), indicating light-induced and phosphorylation-dependent 14-3-3 association to mediate PM release of NPH3. Given that the amphipathic helix localizes ~30–45 residues upstream of the 14-3-3-binding site (Fig. 2a), 14-3-3 binding to NPH3 is expected to induce a substantial conformational change that liberates the amphipathic helix from the PM. The molecular mechanism of NPH3 internalization is hence different from the— likewise PM-associated—photoreceptor phot1, trafficking of which occurs via vesicles through the endosomal recycling pathway[54]. Now, what about the BL-triggered "general" dephosphorylation of NPH3? Based on our findings, this posttranslational modification temporally succeeded light-induced S744 phosphorylation (Fig. 6c). Furthermore, "general" dephosphorylation was coupled to BL-triggered S744 phosphorylation, irrespective of the subcellular localization of NPH3 (Fig. 6a, d). We therefore suspect phosphorylation-dependent 14-3-3

binding to be required for BL-induced "general" dephosphorylation of NPH3 as well—a hypothesis that will be examined by future research.

Re-transfer of BL-irradiated seedlings to darkness triggers (i) dephosphorylation of S744 linked to 14-3-3 dissociation. 14-3-3 release is expected to result in a (re)exposure of the amphipathic helix, which subsequently enables (ii) re-association with the PM and presumably (iii) re-phosphorylation of NPH3, represented by a reduced electrophoretic mobility ("general" re-phosphorylation) (Fig. 6b, c). Intriguingly, neither NPH3 variants that constitutively localize to the PM nor mutant versions constitutively detached from the PM are capable of restoring the severe defect in hypocotyl phototropism in nph3-7. Complementation of the nph3-7 phenotype could exclusively be observed upon expression of NPH3 variants that exhibit a light regime-driven dynamic change in subcellular localization (Fig. 7a–c). In summary, we propose a model where S744 phosphorylation-dependent and 14-3-3-driven cycling of NPH3 between the PM and the cytosol critically determine NPH3 function in mediating phototropic signaling in Arabidopsis (Fig. 7e).

In the past, it has been hypothesized that the light-induced internalization of phot1—first described in 2002[55]—may be coupled to light-triggered re-localization of auxin transporters. Functionality of phot1, however, was unaffected when internalization of the photoreceptor was effectively prevented by PM tethering via lipid anchoring[56]. Altogether, the change in subcellular localization does not seem to be essential for signaling of phot1, but of its downstream signaling component NPH3 (Fig. 7e). Light-induced and 14-3-3-mediated detachment of NPH3 from the PM might hence account for BL-driven changes in PIN polarity required for hypocotyl phototropism. Plant 14-3-3 proteins have been shown to contribute to the subcellular polar localization of PIN auxin efflux carrier and, consequently, auxin transport-dependent growth[7]. NRL proteins in turn act as signal transducers in processes involving auxin (re)distribution in response to developmental or environmental signals[18], hence providing a likely link between 14-3-3 and PIN polarity. One subfamily of the NRL protein family consists of MAB4/ ENP-like (MEL) polypeptides, playing a critical role in auxin-regulated organogenesis in Arabidopsis[57–59]. MEL proteins exhibited a polar localization at the cell periphery, which was almost identical to that of PIN proteins[60,61] and were recently shown to maintain PIN polarity by limiting lateral diffusion[29]. Thus, one attractive hypothesis is that certain NRL proteins contribute either to the maintenance or to a dynamic change of the subcellular polarity of PIN auxin carriers, thereby regulating auxin (re)distribution. Given that several NRL proteins are able to interact with 14-3-3 via a C-terminal binding motif (Supplementary Fig. 5), phosphorylation-dependent 14-3-3 association might constitute a crucial mechanism of regulation for NRL proteins and consequently polarity of PIN proteins.

## Methods

**Plant materials, transformation, and growth conditions.** Arabidopsis thaliana (ecotype Columbia-0 (Col-0)) expressing 14-3-3 epsilon:GFP under control of the native promoter has been described recently[7]. Seeds of A. thaliana nph3-7 (SALK_110039, Col-0 background) were obtained from the Nottingham Arabidopsis Stock Centre. T-DNA insertion was confirmed by genomic PCR analysis and homozygous lines were identified. Stable transformation of nph3-7 followed standard procedures.

Seeds were surface sterilized and planted on solid half-strength Murashige and Skoog (MS) medium (pH 5.8). Following stratification in the dark for 48–72 h at 4 °C, seeds were exposed to fluorescent white light for 4 h. Seedlings were then grown in darkness for 68 h at 20 °C. Subsequently, the etiolated seedlings were either kept in darkness or irradiated with BL (overhead BL (1 μmol m$^{-2}$ s$^{-1}$) for up to 40 min or, alternatively, treatment with the GFP laser (488 nm) for up to 11 min during confocal observation of hypocotyl cells, as specified in the figure legends).

Independent experiments were carried out at least in triplicates. Representative images are presented.

Agrobacterium-mediated transient transformation of 3–4 weeks old N. benthamiana plants was performed as described[62]. Agrobacterium tumefaciens strain GV3101, transformed with the binary vector of interest, was resuspended in infiltration solution (10 mM MES pH 5.6, 10 mM MgCl$_2$, 150 μM acetosyringone) at an OD$_{600}$ of 0.1–0.2 and infiltrated into the abaxial epidermis of N. benthamiana leaves. For co-transformation, a 1 : 1 mixture was used. Freshly transformed tobacco plants were kept under constant light for 24 h, subsequently transferred to darkness for 17 h (dark adaptation), and finally either kept in darkness or irradiated (overhead BL (10 μmol m$^{-2}$ s$^{-1}$) for up to 40 min or, alternatively, treatment with the GFP laser (488 nm) for up to 11 min during confocal inspection of abaxial leaf epidermis cells, as specified in the figure legends). Independent experiments were carried out at least in triplicates. Representative images are presented.

**Cloning procedures.** A 2.1 kb NPH3 promoter fragment was PCR-amplified from Col-0 genomic DNA and the cDNA of NPH3 was amplified from Col-0 cDNA. The respective primers were characterized by BsaI restriction sites allowing for the usage of the Golden Gate-based modular assembly of synthetic genes for transgene expression in plants[63]. Following A-tailing, the individual PCR products were directly ligated into the pGEM-T Easy vector (Promega, Madison, USA), yielding level I vectors LI A-B pNPH3 and LI C-D NPH3, respectively. Golden Gate level II assembly was performed by BsaI cut ligation and by using the modules LI A-B pNPH3, LI B-C GFP[63] or LI B-C mCherry[63], LI C-D NPH3, LI dy D-E[63], LI E-F nos-T[63], and LI F-G Hygro[63]. All plasmids were diluted to a final concentration of 100 ng/μl. In a 15 μl reaction, 1 μl of each plasmid was incubated with 0.5 μl of BsaI (Thermo Scientific, Waltham, USA), 0.75 μl T4 ligase (Thermo Scientific), and 1.5 μl ligase buffer. Reactions were incubated in a thermocycler for 25 cycles, cycling between 37 °C for 2 min and 16 °C for 5 min, followed by 37 °C for 5 min, 50 °C for 5 min, and 80 °C for 5 min. Finally, 0.5 μl T4 ligase was added. Following an incubation at 37 °C for 1 h, a 3 μl aliquot was transformed into Escherichia coli TOP10.

For CoIP of fluorophore-tagged NPH3 and 14-3-3 transiently expressed in N. benthamiana, the corresponding cDNA was cloned into the 2in1 GATEWAY™ compatible vector pFRETcg-2in1-NC[64] via GATEWAY™ technology.

Cloning of N-terminally fluorophore-tagged NPH3 variants (GFP and/or RFP) into the destination vectors pB7WGR2 and/or pH7WGF2[65] for stable or transient overexpression followed standard GATEWAY™ procedures. Transgenic plants were selected based on the hygromycin resistance conferred by pH7WGF2 and homozygous lines were established. The 35S-driven PHOT1:GFP[54] and the 35S::MAP:mCherry:SAC1/SAC1$_{DEAD}$ transformation vectors[33] have been described before, respectively.

Site-directed mutagenesis was performed by PCR. PCR products and products of mutagenesis were verified by sequencing.

A complete list of oligonucleotides used for PCR is provided in Supplementary Table 2.

**Expression and purification of proteins.** For bacterial expression of the Arabidopsis 14-3-3 isoform omega as RGS(His)$_6$-tagged protein, the corresponding cDNA was amplified by PCR and cloned into the expression vector pQE-30 (Qiagen, Hilden, Germany). Transformed E. coli M15 was grown in liquid lysogeny broth (LB) medium containing ampicillin (100 μg/ml) and kanamycin (25 μg/ml) at 37 °C until an OD$_{600}$ of 0.6. Protein expression was induced by adding isopropyl β-d-1-thiogalactopyranoside (IPTG) to a final concentration of 0.5 mM. Following overnight growth at 16 °C, bacteria were collected by centrifugation. The pellet was frozen in liquid nitrogen. Following thawing on ice, the cells were resuspended in lysis buffer (50 mM NaH$_2$PO$_4$, 300 mM NaCl, 10 mM imidazole pH 8.0 using NaOH) containing lysozyme (2 mg/ml). Cells were lysed by sonication. Purification under native conditions was done by using the cleared M15 lysate and Ni$^{2+}$-NTA agarose (Qiagen) according to the manufacturer's protocol.

For bacterial expression of the Arabidopsis NPH3 C-terminal 51 residues fused to glutathione S-transferase (GST), the corresponding cDNA fragment was amplified by PCR and cloned into the GST expression vector pGEX-4T-1 (Cytiva, Marlborough, USA). Transformed E. coli BL21(DE3) was grown in 2 × yeast extract and tryptone (YT) medium containing ampicillin (100 μg/ml) at 37 °C until an OD$_{600}$ of 0.6. Protein expression was induced by adding IPTG to a final concentration of 0.1 mM. Following overnight growth at 20 °C, bacteria were collected by centrifugation. The pellet was frozen in liquid nitrogen. Following thawing on ice, the cells were resuspended in bacterial protein extraction reagent B-PER (5 ml/g fresh weight) (Thermo Scientific). GST fusion proteins were purified from the cleared bacterial lysate using GSH-Sepharose 4 Fast Flow equilibrated with phosphate-buffered saline (PBS) according to the manufacturer's protocol (Cytiva). Elution of bound proteins was achieved by adding 10 mM reduced glutathione in 50 mM Tris pH 8.0. Free GST protein was expressed and purified to serve as a negative control.

**Cell-free protein expression.** Reactions were performed using the TNT® T7 Quick Coupled Transcription/Translation System (Promega) with 1 μg of vector (NPH3 or variants in pGADT7), 40 μl TNT® Quick Master Mix, and 1 μl 1 mM

methionine in a total volume of 50 µl. Protein expression was carried out at 30 °C for 90 min. Immunodetection was performed by using an anti-hemagglutinin (HA) antibody (HA-tag encoded by pGADT7).

**Preparation of microsomal membranes**. Microsomal membrane fractions were prepared from transiently transformed *N. benthamiana* leaves. Tissue was homogenized with 3 ml homogenization buffer per gram fresh weight (50 mM Hepes pH 7.8, 500 mM sucrose, 1% PVP-40, 3 mM dithiothreitol (DTT), 3 mM EDTA, supplemented with Complete Protease Inhibitor Mixture (Roche, Basel, Switzerland) and Phosphatase Inhibitor Mix 1 (Serva, Heidelberg, Germany)). The homogenate was centrifuged at $10,000 \times g$ for 20 min at 4 °C. The supernatant was filtered through MiraCloth and subsequently centrifuged at $100,000 \times g$ for 45 min at 4 °C. The microsomal pellet was resuspended in 5 mM Tris/MES pH 6.5, 330 mM sucrose, 2 mM DTT, supplemented with Complete Protease Inhibitor Mixture (Roche) and Phosphatase Inhibitor Mix 1 (Serva).

**Phospholipid-binding assays**. For lipid binding assays, either NPH3 variants expressed in a cell-free system or purified recombinant GST fusion proteins were applied. Lipid overlay assays using phosphorylated derivatives of phosphatidylinositol (PIP) strips were performed following the manufacturer's instructions (Echelon, Salt Lake City, USA). In brief, membranes were blocked overnight at 4 °C in blocking buffer (4% fatty acid-free bovine serum albumin in PBS-T (0.1% Tween-20 in PBS)). Purified proteins (0.1 µg/ml blocking buffer) or 10–50 µl of the cell-free expression reaction (volume adjusted according to prior immunodetection of individual reactions) were incubated with PIP-strip membranes for 1 h at room temperature and washed three times for 10 min with PBS-T. Subsequently, bound proteins were visualized by immunodetection of either GST (GST fusion proteins) or the HA-tag (cell-free expression).

Liposome-binding assays were conducted essentially as described by ref. [66] with slight modifications. All lipids were obtained from Avanti Polar Lipids (Birmingham, USA). Liposomes were prepared from 400 nmol of total lipids at the molar ratios: PC : PE, 1 : 1; PC : PE : PI4P, 2 : 2 : 1; PC : PE : PA, 2 : 2 : 1 by using a mini-extruder (Avanti Polar Lipids) at room temperature. Following centrifugation at $50,000 \times g$ for 15 min at 22°, the liposome pellets were resuspended in 25 µl binding buffer (150 mM KCl, 25 mM Tris-HCl pH 7.5, 1 mM DTT, 0.5 mM EDTA, supplemented with Complete Protease Inhibitor Mixture (Roche)). Purified GST-NPH3-C51 variants in binding buffer were centrifuged at $50,000 \times g$ to get rid of any possible precipitates. Subsequent to an incubation of liposomes and proteins (500 ng in 25 µl binding buffer) on an orbital shaker platform for 45 min, the samples were centrifuged at $16,000 \times g$ for 30 min at room temperature. The liposome pellet was washed twice with binding buffer. Liposome-bound GST-NPH3-C51 variants were detected by immunoblotting with anti-GST antibodies.

**Y2H, SDS-PAGE, and western blotting**. For yeast two-hybrid analyses, the individual constructs were cloned into the vectors pGADT7 and pGBKT7 (Takara Bio, Kusatsu, Japan), and co-transformed into the yeast strain PJ69-4A. Activity of the *ADE2* reporter was analyzed by growth of co-transformed yeast on synthetic dropout (SD) medium lacking adenine.

SDS-PAGE, western blotting, and immunodetection followed standard procedures. Total proteins were prepared from 3-day-old etiolated *Arabidopsis* seedlings (50 seedlings) or transiently transformed *N. benthamiana* leaves (2 leaf discs) by directly grinding in 100 µl $2 \times$ SDS sample buffer under red safe light illumination. Chemiluminescence detection was performed with an Amersham Image Quant800 (Cytiva) system.

The rabbit anti-NPH3-S744P (dilution used: 1 : 500) and anti-NPH3 (dilution used: 1 : 1000) antibodies were generated with the phosphorylated synthetic peptide $NH_2$-PPRKPRRWRN-S($PO_3H_2$)-IS-COOH followed by affinity purifications against the non-phosphorylated and phosphorylated peptide, respectively, at Eurogentec (Liege, Belgium).

In addition, the following antibodies were used in this study: anti-GST (1 : 2000, Cytiva, catalog number: 27457701, lot: 5205496), anti-HA high affinity (1 : 2000, Roche, catalog number: 11867423001, lot unknown), anti-GFP rabbit IgG fraction (1 : 1000, Thermo Scientific, catalog number: A-11122, lot: 2180255), anti-RFP 5F8 (1 : 1000, ChromoTek, Planegg-Martinsried, Germany, catalog number: 5F8-100, lot: 90228002AB-10), anti-RGS-His antibody (1 : 2000, Qiagen, catalog number: 34650, lot: 163033250).

**CoIP and MS analysis**. *Arabidopsis* seeds expressing 14-3-3 epsilon-GFP (endogenous promoter) and, as control, GFP (UBQ10 promoter) were sown on half-strength MS plates and grown in the dark for 3 days. Subsequently, the etiolated seedlings were either kept in darkness or treated with overhead BL (1 µmol $m^{-2} s^{-1}$) for 30 min. Three grams of plant tissue were used under red safe light illumination for immunoprecipitation essentially as described in[67] with slight modifications. The seedlings were ground thoroughly in liquid nitrogen and suspended in lysis buffer (50 mM Tris pH 7.5, 150 mM NaCl, supplemented with Complete Protease Inhibitor Mixture (Roche) and PhosSTOP phosphatase inhibitor cocktail (Roche)) containing 1% Triton X-100. After 30 min incubation on ice, cell debris-removed supernatants were incubated with 50 µl GFP-Trap beads (ChromoTek) for 3 h in the cold room with mild rotation. The beads were washed three times with lysis buffer containing

0.1% Triton X-100, followed by washing with lysis buffer. The final precipitate in Laemmli buffer was analyzed by MS at the Proteome Center Tübingen, University of Tübingen. Following a tryptic in gel digestion, liquid chromatography-MS/MS analysis was performed on a Proxeon Easy-nLC coupled to an QExactiveHF mass spectrometer (method: 60 min, Top7, higher-energy C-trap dissociation (HCD)). Processing of the data was conducted using MaxQuant software (vs 1.5.2.8). The spectra were searched against an *A. thaliana* UniProt database. Raw data processing was done with 1% false discovery rate setting. Two individual biological replicates were performed and proteins identified in only one of the two experiments were omitted from the list of 14-3-3 epsilon-GFP interaction partners. Protein signal intensities of well-known 14-3-3 client proteins (Fig. 3c) were converted to normalized abundance of the bait protein. Fold changes in relative abundance of BL treatment vs. darkness (BL vs. D) were calculated (Supplementary Table 1).

*Arabidopsis nph3-7* ectopically expressing GFP:NPH3 and *N. benthamiana* leaves transiently overexpressing fluorophore-tagged proteins were immunoprecipitated under red safe light illumination according to ref. [68]. Growth and light irradiation of the plants is specified in the figure legends. Immunoprecipitations were performed with 50 mg of tissue. Ground material was resuspended in solubilization buffer (25 mM Tris pH 8.0, 150 mM NaCl, 1% NP40, 0.5% sodium deoxycholate, supplemented with Complete Protease Inhibitor Mixture (Roche) and Phosphatase Inhibitor Mix 1 (Serva)). After 1 h incubation in the cold room with mild rotation, cell debris-removed supernatants were incubated with 20 µl GFP-Trap beads (ChromoTek) for 1 h in the cold room with overhead rotation. The beads were washed twice with solubilization buffer, followed by two washing steps with 25 mM Tris pH 8.0, 150 mM NaCl. Protein blottings were probed directly with an appropriate antibody or, alternatively, used for 14-3-3 far-western analysis.

In vivo interaction of phot1:GFP and N-terminally RFP-tagged NPH3 variants was tested by using solubilized microsomal proteins obtained from dark-adapted *N. benthamiana* plants ectopically co-expressing the proteins of interest. Solubilization was achieved by adding 0.5% Triton X-100 to resuspended microsomal proteins by centrifugation at $50,000 \times g$ for 30 min at 4 °C. The supernatant was added to GFP-Trap Beads (ChromoTek) and incubated at 4 °C for 1 h. Precipitated beads were washed six times with 50 mM HEPES pH 7.8, 150 mM NaCl, 0.2% Triton X-100. Finally, proteins were eluted by SDS sample buffer and separated by SDS-PAGE.

**14-3-3 far-western analysis**. Anti-GFP immunoprecipitates obtained from *Arabidopsis nph3-7* stably overexpressing GFP:NPH3 were separated by SDS-PAGE and transferred to nitrocellulose. Nonspecific sites were blocked by incubation with 4% (w/v) milk powder in Tris-buffered saline (TBS) at room temperature for at least 1 h. Subsequently, the membrane was incubated overnight at 4 °C (followed by 1 h at room temperature) with purified recombinant RGS(His)$_6$-tagged 14-3-3 isoform omega diluted to 20 µg $ml^{-1}$ in 50 mM MOPS pH 6.5, 20% glycerol, 5 mM $MgCl_2$, and 2 mM DTT. After washing with TBS, immunodetection of RGS(His)$_6$-tagged 14-3-3 was performed by applying the anti-RGS(His)$_6$ antibody (Qiagen) in combination with a secondary anti-mouse horseradish peroxidase antibody.

**Hypocotyl phototropism analysis**. *A. thaliana* seedlings were grown in the dark on vertically oriented half-strength MS plates for 48 h. Etiolated seedlings were then transferred to an light emitting diode (LED) chamber and illuminated with unilateral BL (1 µmol $m^{-2} s^{-1}$) for 24 h. Plates were scanned and the inner hypocotyl angle was measured for each seedling using ImageJ[69]. The curvature angle was calculated as the difference between 180° and the measured value. For each transgenic line, three biological replicates ($n \geq 30$ seedlings per experiment) were performed, alongside with the appropriate controls (Col-0, *nph3-7*) (see Source Data file, phototropism sheet I to VI).

**Confocal microscopy**. Live-cell imaging was performed using the Leica TCS SP8 (upright) confocal laser scanning microscope. Imaging was done by using a × 63/ 1.20 water-immersion objective. For excitation and emission of fluorophores, the following laser settings were used: GFP, excitation 488 nm, emission 505–530 nm; RFP, excitation 558 nm, emission 600–630 nm. All confocal laser scanning fluorescence microscopy (CLSM) images in a single experiment were captured with the same settings using the Leica Confocal Software. All experiments were repeated at least three times. Images were processed using LAS X light (version 3.3.0.16799).

Single-cell time-lapse imaging was carried out on live leaf tissue samples from *N. benthamiana* transiently expressing RFP:NPH3. PM detachment was induced by means of the GFP laser (488 nm) and image acquisition (RFP-laser) was done for the duration of 32 min by scanning 30 consecutive planes along the Z axis covering the entire thickness of an epidermal cell. Z-projection was done for each 3.5 min interval. Five biological replicates were performed.

**Data analysis**. The angles of phototropic curvature of all analyzed *Arabidopsis* genotypes ($\geq 30$ seedlings per genotype) were measured using ImageJ software and analyzed using Graphpad Prism software. Statistical significance of the data was assessed using one-way analysis of variance (ANOVA) followed by post hoc Tukey's multiple comparison test ($P < 0.05$). Error bars represent standard deviation. The results of all three biological replicates and detail of one-way ANOVA are provided in the Source Data file, phototropism sheet I to VI. For IP-MS data,

Student's *t*-tests were performed using MS-Excel. For all image quantifications related to single-cell time-lapse imaging, randomly sampled unsaturated confocal images (1024 × 1024 pixels, 246 × 246 μm) were used with an image analysis protocol implemented in the ImageJ software[69] as described[70]. A random image was selected from the data set and parameters such as local threshold, background noise, object size, and shape were determined. The obtained parameters were used for image analysis of the whole data set following exactly the published step-by-step protocol[70]. Unless otherwise stated, graphs present data from a single experiment.

**Reporting summary**. Further information on research design is available in the Nature Research Reporting Summary linked to this article.

## Data availability

All data are available within this article and its Supplementary Information. The mass spectrometry proteomics data have been deposited to the ProteomeXchange Consortium via the PRIDE[71] partner repository with the data set identifier PXD028530 (http://www.ebi.ac.uk/pride/archive/projects/PXD028530). Source data are provided with this paper.

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

## Acknowledgements
This work was supported by the German Research Foundation (DFG) with a grant to C.O. (CRC 1101-B09) and grants for scientific equipment (INST 37/819-1 FUGG, INST 37/991-1 FUGG). We are grateful to Yvon Jaillais and John M. Christie for providing the constructs 35S::MAP:mCherry:SAC1/SAC1$_{DEAD}$ and 35S::PHOT1:GFP, respectively. We further thank John M. Christie for sharing data and stimulating discussions, Sandra Richter for SP8 support, Nathalie Zgoda for data analysis, and Marissa Glauner and Marius Müller for assistance with phospholipid-binding assays. MS analysis was done at the Proteome Center Tübingen, University of Tübingen, and we thank Professor Boris Macek for valuable instructions.

## Author contributions
C.O. conceived the project. L.R., T.S., P.M., C.T., and C.O. designed experiments. L.R., T.S., P.M., C.T., J.K., A.B., and I.D.-B. performed experiments. L.R., T.S., P.M., C.T., I.D.-B., and C.O. analyzed data. The manuscript was principally written by C.O. with comments from all authors.

## Funding

## Competing interests
The authors declare no competing interests.
