## [Peer Review File · Nature Communications]

Light-triggered and phosphorylation-dependent 14-3-3 association with NON-PHOTOTROPIC HYPOCOTYL 3 is required for hypocotyl phototropismREVIEWER COMMENTS

Reviewer #1 (Remarks to the Author):

The paper by Reuter et al. presents impressive work aimed at the characterization of NPH3 functioning in seedlings. Two major findings, the nature of NPH3 interaction with the plasma membrane and light-dependent phosphorylation of NPH3 followed by 14-3-3 binding are of primary interest to the researcher community.

Microscopic observations are of primary importance for conclusions raised in this paper. However, some panels lack images showing the localization patterns of wild-type NPH3 in particular experiments. This is important as transient transformation of *N. benthamiana* is sensitive to the state of plants and thus the control line should be always included as a reference to show that the observed differences in localization stem from mutagenesis of the protein itself. I would also like to ask for three biological replicates of physiological assays using the NPH3 expressing lines.

Before I recommend this paper for publishing, I would like to ask for a rearrangement of figure panels. The story presented in the paper is consistent, however extremely difficult to follow, due to figure panels inconsistency. It very inconvenient to follow the data presented in several figures simultaneously. The figures should be prepared in such a way that the reader can refer to them in a sequence and does not need to jump between their fragments. Figure numbering should in general match the text of the manuscript. Also, figure legends are not clear enough to me, as the description tends to merge quite different experiments. Descriptions should also refer in detail to what is presented on figures and be self-explanatory. The word antepenultimate is not intuitive and I would like to ask for a different expression.

Detailed comments:

Fig.1,A,D Please provide control images.

Fig 1B. Please indicate GST: NPH3-S774D in the figure legend.

Line 115. Please refer in the full name to the analyzed fusion proteins for example GST: NPH3-C51 so it is easier to refer to the figures.

Line 140. Were the mutations in the replacement variants of NPH3 introduced into the whole mutated proteins or only the C51 fragment? Please clarify the description in the text and the figure legend.

Line 146. Please refer precisely to muteins used.

Fig.2B. Were the proteins tagged with GST (figure legend) or HA (marked on the figure)?

Fig.2D. Please indicate the localization of wild-type NPH3 in the appropriate control lines. I do not like the idea of comparing the localization of proteins fused with different fluorescent protein tags, as in our hands GFP and RFP show different expression levels and sensitivity to photobleaching which affects the signal intensities. I am afraid that comparing the localization of proteins fused with different fluorescent protein tags may lead to biased assessments.

Line 159. Why interactions with omega 14-3-3 are shown in figure 3A, but 14-3-3 epsilon: GFP is further analyzed by MS? The IP experiments are done for the 14-3-3 omega, but this is not indicated in the text

Fig.3B, D. Please indicate the type of 14-3-3 in the figure and in the text.

Line 190. Please comment on why phosphomimic variants do not allow for 14-3-3 binding.

Line 219-220. Please rephrase this sentence.

Line 224. Please indicate the wavelength for the GFP laser.

Line 269- 270. Please indicate the reference.

Fig. 4A. Please provide data from 3 biological replicates. Similarly for Fig.S3A.

Fig. 4F, G. Please indicate the number of analyzed cells.

Fig.6B. Please add GFP-NPH3 localization in the control line.

Fig.6D. Please indicate what unbound/bound means in the figure legend.

Line 470-471. Please describe the treatments in the M and M sections as well.

Fig. S2. For colocalization studies please provide images also from separate channels.

Reviewer #2 (Remarks to the Author):

The authors present truly outstanding results. Their work revolutionises our current understanding of the molecular basis for phototropism. They present results that not only solve the long standing mystery of how NPH3 attaches to membranes, but go on to show in exquisite detail how NPH3-membrane localisation is regulated and how this impacts on NPH3 function. This MS will have a profound impact on the field. The MS is already looking great, but as a reviewer it is my role to point out some areas that I think could be improved and so I detail these below.

The methodology in the paper is innovative and yet sound. In the methods section, several techniques are cited, rather than described. To ensure reproducibility, please describe these methods. In some instances, inappropriate statistical tests are used. The statistical tests used do not correct for multiple testing and so other tests (e.g. 1-way ANOVA with a post hoc test) may be more appropriate. Some continuous data are presented as histograms, where violin/ box plots would give the reader a better understanding of the variation in these experiments.

Most of the conclusions in the MS are completely well founded. I did however find a few areas where language could be tightened or a more appropriate control could have been used:

Figure 2D: Why did the authors use different FP for these experiments? As the authors note, laser excitation of GFP also effects NPH3 localisation. To make comparisons between the localisation of these truncated proteins it would be beneficial to use the same FP.

Fig 3A. It's interesting that the binding of 14:3:3 to NPH3 also occurs in yeast. Do the authors have any suggestions of how plant-specific NPH3 (presumably) becomes phosphorylated in the absence of phot1?

Line 311: "General' dephosphorylation of NPH3 is thus not coupled to PM dissociation. Moreover, it is neither a prerequisite nor a consequence of condensate assembly, rather it seems to require prior light-triggered and S744 phosphorylation-dependent 14-3-3 association". While I am sympathetic to the hypothesis put forward by the authors, strictly speaking they do not show that association of 14:3:3 is required for NPH3 de-phosphorylation. Rather they show that BL-dependent phosphorylation of S744 is required for general de-phosphorylation AND for the association of 14:3:3. If they want to make the current conclusion they need to show that general de-phos does not occur in the absence of 14:3:3s. Given the highly redundant function of 14:3:3s, this is no small task. The easiest solution would be to moderate this conclusion (also for the statements in line 347, 399 and the general conclusions).

Figure 6C. GFP:NPH3-4K/A appears to run on the gel as a 'general dephosphorylated' NPH3. Does that not imply that cytosolic localisation is sufficient for general dephosphorylation? How does this square with the results presented in Figure 5D?

Line 377: I don't follow how the residual function of NPH3-S744A can be explained by redundancy with RPT2. The NPH3-S744A construct rescues phototropism in the *nph3-7* background. The most simple explanation is that NPH3-S744A still retains some function.

In addition to these remarks, I also had a few suggestions for where I think that the MS could be modified to enhance fullness and readability:

Line 42: The authors should cite studies showing phot1 transphosphorylation ie. BLUS1, CBC1, ABCD19 and PSK4

Line 119: I this is referring to RFP-constructs (1D) and GFP constructs (S2A)?

The figure legends could be simplified. Currently, several sub-figures are referred to within the same heading. For example: “(Figure 3. B, D) In vivo interaction of mCherry:NPH3 variants and 14-3-3 omega:mEGFP in transiently transformed *N. benthamiana* leaves. Expression of transgenes was driven by the 35S promoter. Freshly transformed tobacco plants were either kept under constant light for 42 h (B) or kept under constant light for 24 h and subsequently transferred to darkness for 17h with (BL) or without (D) blue light treatment ($5 \mu\text{mol m}^{-2} \text{sec}^{-1}$ 912) for the last 40 minutes (D)”. It would be a lot easier for the reader to follow if the legend for B and D was separated. This will inevitably require some repetition, but the legend will become much clearer as a result.

Line 179: mostly

Line 200: suggesting (indicating is too strong)

Line 305: neither NPH3 variant?

It took me a little while to understand summary figure 6E. It might help to clearly separate the 3 panels and to label them as wild type NPH3 and constitutive membrane-bound / unbound variants.

Line 397: ‘also in plants’ is not necessary (any more than also in bacteria, also in animals etc)!

Best wishes,

Scott Hayes

Response to Reviewers:

We would like to thank the Referees for constructive comments that helped us to improve our MS. Below are point-by-point replies to all Referee queries and explanations for changes we have made to the original MS. To facilitate assessment, we have copied reviewer's queries (Q) ahead of our replies (R):

Reply to comments of Referee 1:

Q1: Microscopic observations are of primary importance for conclusions raised in this paper. However, some panels lack images showing the localization patterns of wild-type NPH3 in particular experiments. This is important as transient transformation of *N. benthamiana* is sensitive to the state of plants and thus the control line should be always included as a reference to show that the observed differences in localization stem from mutagenesis of the protein itself.

R1: We now included images showing the localization pattern of the respective NPH3 control in all figure panels displaying CLSM analyses (Fig 1, 2, 4, 5, 7, Supplementary Fig. 2, 3, 4).

Q2: I would also like to ask for three biological replicates of physiological assays using the NPH3 expressing lines.

R2: We performed a third biological replicate of the analysis of the phototropic response in all control and transgenic lines expressing GFP:NPH3 variants under control of the 35S promoter (endogenous promoter see below). The data as well as the data analysis are available in a separate source data Excel document referred to as 'source data phototropism' (sheet I to III: 35S *nph3-7* phototropism). We, however, faced a germination problem with *nph3-7* line #5 expressing GFP:NPH3-S744A under control of the endogenous promoter (see previous Fig. S3 A). We therefore have chosen another transgenic *nph3-7* line expressing *pNPH3::GFP:NPH3-S744A* (# 13, see Supplementary Fig. 4 in the revised manuscript) for further analysis. The subcellular localization of the transgene both in darkness and upon blue light treatment was examined. Furthermore, three biological replicates with the transgenic *nph3-7* lines expressing pNPH3 driven constructs, alongside with the controls were performed. The data as well as the data analysis are available in the abovementioned source data Excel document (sheet IV to VI: pNPH3 *nph3-7* phototropism).

Q3: Before I recommend this paper for publishing, I would like to ask for a rearrangement of figure panels. The story presented in the paper is consistent, however extremely difficult to follow, due to figure panels inconsistency. It very inconvenient to follow the data presented in several figures simultaneously. The figures should be prepared in such a way that the reader can refer to them in a sequence and does not need to jump between their fragments. Figure numbering should in general match the text of the manuscript.

R3: Figures and figure panels were completely rearranged so that Figure numbering matches the text of the main manuscript.

Q4: Also, figure legends are not clear enough to me, as the description tends to merge quite different experiments. Descriptions should also refer in detail to what is presented on figures and be self-explanatory.

R4: Figure legends were separated and simplified, including enough information to understand the figure without referring to the main text.

Q5: The word antepenultimate is not intuitive and I would like to ask for a different expression.

R5: The word antepenultimate was replaced by 'third last'.

Detailed comments Referee 1:

Q6: Fig.1,A,D Please provide control images.

R6: Control images are now shown in Fig. 1a and Fig. 1d (see **R1**).

Q7: Fig 1B. Please indicate GST: NPH3-S774D in the figure legend.

R7: Due to figure rearrangement (see **R3**) the lipid overlay assay performed with GST:NPH3-C51-S774D is now shown in Fig. 4c. The analyzed protein is mentioned in the figure legend.

Q8: Line 115. Please refer in the full name to the analyzed fusion proteins for example GST: NPH3-C51 so it is easier to refer to the figures.

R8: In all cases, we now referred to the full name of the studied fusion proteins. For instance, GST:NPH3-C51 is mentioned in line 121.

Q9: Line 140. Were the mutations in the replacement variants of NPH3 introduced into the whole mutated proteins or only the C51 fragment? Please clarify the description in the text and the figure legend.

Line 146. Please refer precisely to muteins used.

Fig.2B. Were the proteins tagged with GST (figure legend) or HA (marked on the figure)?

R9: Thanks for pointing this out. We realized that the previous figure legend 2(B) was not correct. In fact, the mutations were introduced in both the HA-tagged full-length NPH3 protein and the GST-NPH3-C51 fusion protein. While the first-mentioned protein was used for lipid overlay assays (Fig. 2b), liposome binding assays were performed by using the GST-NPH3-C51 variants (Fig. 2c). This is now clearly indicated in the main text (line 146) as well as the legend of Fig. 2.

Q10: Fig.2D. Please indicate the localization of wild-type NPH3 in the appropriate control lines. I do not like the idea of comparing the localization of proteins fused with different fluorescent protein tags, as in our hands GFP and RFP show different expression levels and sensitivity to photobleaching which affects the signal intensities. I am afraid that comparing the localization of proteins fused with different fluorescent protein tags may lead to biased assessments.

R10: As already mentioned (**R1**) we now included images showing the localization pattern of the respective NPH3 control in all figure panels displaying CLSM analyses. Furthermore, we analyzed the localization of 35S::RFP:NPH3-5KR/A in darkness – a representative image is now shown for comparison with other RFP-tagged NPH3 variants in Fig. 2d.

Q11: Line 159. Why interactions with omega 14-3-3 are shown in figure 3A, but 14-3-3 epsilon: GFP is further analyzed by MS? The IP experiments are done for the 14-3-3 omega, but this is not indicated in the text Fig.3B, D. Please indicate the type of 14-3-3 in the figure and in the text.

R11: We now included yeast two hybrid assays performed with the 14-3-3 isoform epsilon (belonging to the epsilon group) to demonstrate that members of both phylogenetic 14-3-3 groups (isoform omega belongs to the non-epsilon group, Fig. 3a, Supplementary Fig. 5a) interact with NPH3 (Supplementary Fig. 4a) as well as other NRL proteins (Supplementary Fig. 5b). In each case, the type of 14-3-3 isoform is indicated in the figure and in the text. Within the main text, we inserted the following statement (lines 170-174): 'A yeast two hybrid screen performed in our lab (see ³⁹) identified NPH3 as putative interactor of several Arabidopsis 14-3-3 isoforms, among those representatives of both phylogenetic 14-3-3 groups, the non-epsilon group (isoform omega, Fig. 3a) and the epsilon group (isoform epsilon, Supplementary Fig. 4a) ⁷. In contrast to phot1 ⁸ 14-3-3 isoform specificity was thus not observed for binding to NPH3.'

Q12: Line 190. Please comment on why phosphomimic variants do not allow for 14-3-3 binding.

R12: The requested information has been added (lines 204-207): 'Phosphomimic variants (NPH3-S744D/S744E), however, do not allow for 14-3-3 binding (Fig. 3a, Supplementary Fig. 4a), consistent with the general finding that aspartate and glutamate do not provide good phosphomimetic residues with respect to 14-3-3 binding ⁴⁵.'

Q13: Line 219-220. Please rephrase this sentence.

R13: The sentence was rephrased (lines 238-241): 'Altogether, the C-terminal domain serves a dual function in determining the subcellular localization of NPH3 since it comprises both the amphipathic helix required for phospholipid-dependent PM association in darkness and the 14-3-3 binding motif mediating BL-triggered PM dissociation.'

Q14: Line 224. Please indicate the wavelength for the GFP laser.

R14: The wavelength is indicated (line 244-246).

Q15: Line 269- 270. Please indicate the reference.

R15: The reference is indicated (line 295).

Q16: Fig. 4A. Please provide data from 3 biological replicates. Similarly for Fig.S3A.

R16: We performed three biological replicates for the analysis of the phototropic response (see **R2**). The graphs shown in Fig. 4a, Fig. 7a (previous Fig. 6A) and Supplementary Fig. 4b represent data from a single experiment. Source data for all biological replicates including data analysis are provided with this paper (separate source data Excel document 'source data phototropism', see **R2**).

Q17: Fig. 4F, G. Please indicate the number of analyzed cells.

R17: As (already) indicated in the figure legend (now Fig. 5 d-f) a single cell time lapse imaging of RFP:NPH3 condensation was performed. This single cell experiment was repeated five times and is described in the 'Material and Methods' section as follows (lines 706-710): 'PM-detachment was induced by means of the GFP-laser (488 nm) and image acquisition (RFP-laser) was done for the duration of 32 min by scanning 30 consecutive planes along the Z axis covering the entire thickness of an epidermal cell. Z-projection was done for each 3.5 min interval. Five replicates were performed.'

Q18: Fig.6B. Please add GFP-NPH3 localization in the control line.

R18: GFP:NPH3 localization in the control line has been added (now Fig. 7 b) (see **R1**).

Q19: Fig.6D. Please indicate what unbound/bound means in the figure legend.

R19: We exchanged unbound/bound for flowthrough/precipitate (IP) and indicated the meaning in the figure legend (now Fig. 7d).

Q20: Line 470-471. Please describe the treatments in the M and M sections as well.

R20: The light treatments are now described in the M and M section as well (lines 500-503 for transgenic Arabidopsis seedlings, lines 510-514 for *N. benthamiana*).

Q21: Fig. S2. For colocalization studies please provide images also from separate channels.

R21: The colocalization studies shown in the previous Fig. S2 are now available in Supplementary Fig. 3, alongside with the separate channels.

Reply to comments of Referee 2:

Q1: The methodology in the paper is innovative and yet sound. In the methods section, several techniques are cited, rather than described. To ensure reproducibility, please describe these methods.

R1: We now described the majority of methods in detail. In rare cases we, however, precisely followed a step-by-step protocol (i.e. data analysis of single cell time lapse imaging). On that condition, we just cited the corresponding protocol.

Q2: In some instances, inappropriate statistical tests are used. The statistical tests used do not correct for multiple testing and so other tests (e.g. 1-way ANOVA with a post hoc test) may be more appropriate. Some continuous data are presented as histograms, where violin/ box plots would give the reader a better understanding of the variation in these experiments.

R2: With respect to the analysis of the phototropic response (Fig. 4a, Fig. 7a, Supplementary Fig. 4b in the revised manuscript) we now performed one-way ANOVA with Tukey's *post hoc* test and presented the data as box plots. The data as well as the data analysis are available in a separate source data Excel document referred to as 'source data phototropism'.

Q3: Most of the conclusions in the MS are completely well founded. I did however find a few areas where language could be tightened or a more appropriate control could have been used: Figure 2D: Why did the authors use different FP for these experiments? As the authors note, laser excitation of GFP also effects NPH3 localisation. To make comparisons between the localisation of these truncated proteins it would be beneficial to use the same FP.

R3: We completely agree and thus analyzed the localization of 35S::RFP:NPH3-5KR/A in darkness – a representative image is now shown for comparison with other RFP-tagged NPH3 variants in Fig. 2d.

Q4: Fig 3A. It's interesting that the binding of 14:3:3 to NPH3 also occurs in yeast. Do the authors have any suggestions of how plant-specific NPH3 (presumably) becomes phosphorylated in the absence of phot1?

R4: Beyond any doubt, this is an interesting observation and here, we just can speculate. We now discuss this issue as follows (lines 207-211): 'Considering that constitutive 14-3-3 complex formation of other plant targets characterized by a C-terminal binding site, such as the H⁺-ATPase or the transcription factor FD, has been observed in yeast^{46, 47}, light-independent NPH3:14-3-3 interaction in yeast (Fig. 3a, Supplementary Fig. 4a) might arise from a promiscuous kinase with a certain preference for terminal motifs.'

Q5: Line 311: "General' dephosphorylation of NPH3 is thus not coupled to PM dissociation. Moreover, it is neither a prerequisite nor a consequence of condensate assembly, rather it seems to require prior light-triggered and S744 phosphorylation-dependent 14-3-3 association". While I am sympathetic to the hypothesis put forward by the authors, strictly speaking they do not show that association of 14:3:3 is required for NPH3 de-phosphorylation. Rather they show that BL-dependent phosphorylation of S744 is required for general de-phosphorylation AND for the association of 14:3:3. If they want to make the current conclusion they need to show that general de-phos does not occur in the absence of 14:3:3s. Given the highly redundant function of 14:3:3s, this is no small task. The easiest solution would be to moderate this conclusion (also for the statements in line 347, 399 and the general conclusions).

R5: Yes, from a strict point of view, this is absolutely correct! We therefore moderated our conclusion. Lines 334-340: 'Moreover, it is neither a prerequisite nor a consequence of condensate assembly, rather it requires prior light-triggered S744 phosphorylation and potentially 14-3-3 association (Fig. 6a, d). Taken together, we suggest that BL-induced phosphorylation of S744 provokes (i) 14-3-3 association which releases NPH3 from the PM into the cytosol and (ii) 'general' dephosphorylation of NPH3'.

Q6: Figure 6C. GFP:NPH3-4K/A appears to run on the gel as a 'general dephosphorylated' NPH3. Does that not imply that cytosolic localisation is sufficient for general dephosphorylation? How does this square with the results presented in Figure 5D?

R6: This observation puzzled us as well. Nevertheless, provided that 'general dephosphorylation' is not coupled to PM dissociation (**Q5**, see above) there is another explanation for the electrophoretic mobility of this NPH3 mutant that most likely is incapable of PM association following translation: 'general' phosphorylation of NPH3 might take place at the PM! This possibility is now mentioned in lines 350-353: 'Worth mentioning, the electrophoretic mobility of GFP:NPH3-4K/A corresponded to the dephosphorylated version of NPH3 and was not modified by light treatment (Fig. 7c), suggesting that 'general' phosphorylation of NPH3 might take place at the PM.'

Q7: Line 377: I don't follow how the residual function of NPH3-S744A can be explained by redundancy with RPT2. The NPH3-S744A construct rescues phototropism in the *nph3-7* background. The most simple explanation is that NPH3-S744A still retains some function.

R7: I am sorry for this! Of course, residual activity of NPH3-S744A cannot be explained by redundancy with RPT2! We, however, have to keep in mind, that NPH3-S744A is permanently present at the PM. Therefore, the possibility exists that it functions together with RPT2 or other NRL family members. We now rephrased the sentence (lines 406-408): 'Residual functionality might be due to co-action of this constitutively PM-associated NPH3 mutant with certain members of the NRL protein family.'

Q8: In addition to these remarks, I also had a few suggestions for where I think that the MS could be modified to enhance fullness and readability:

Line 42: The authors should cite studies showing phot1 transphosphorylation ie. BLUS1, CBC1, ABCD19 and PSK4

R8: The phot1 substrates known so far are now mentioned in the Introduction (lines 42-46).

Q9: Line 119: I this is referring to RFP-constructs (1D) and GFP constructs (S2A)?

R9: In fact, the (previously misleading) statement should refer to both constructs. This has been rephrased (lines 127-129): 'As expected, transient expression of RFP/GFP:NPH3 Δ C51 in *N. benthamiana* (35S or native promoter) revealed loss of PM recruitment in the dark, as evident by the presence of discrete bodies in the cytosol (Fig. 1d, Supplementary Fig. 2c).

Q10: The figure legends could be simplified. Currently, several sub-figures are referred to within the same heading. For example: "(Figure 3. B, D) In vivo interaction of mCherry:NPH3 variants and 14-3-3 omega:mEGFP in transiently transformed *N. benthamiana* leaves. Expression of transgenes was driven by the 35S promoter. Freshly transformed tobacco plants were either kept under constant light for 42 h (B) or kept under constant light for 24 h and subsequently transferred to darkness for 17h with (BL) or without (D) blue light treatment (5 μ mol m⁻² sec⁻¹ 912) for the last 40 minutes (D)". It would be a lot easier for the reader to follow if the legend for B and D was separated. This will inevitably require some repetition, but the legend will become much clearer as a result.

R10: Figure legends were separated and simplified, including enough information to understand the figure without referring to the main text.

Q11: Line 179: mostly

R11: This has been modified (line 193).

Q12: Line 200: suggesting (indicating is too strong)

R12: 'Indicating' has been exchanged for 'suggesting' (line 220).

Q13: Line 305: neither NPH3 variant?

R13: NO, definitively not! To avoid confusion the sentence was rephrased (line 330): 'Despite the fact that **both** NPH3 variants constitutively localized to cytosolic condensates (Fig. 1a, Fig. 5c).....'

Q14: It took me a little while to understand summary figure 6E. It might help to clearly separate the 3 panels and to label them as wild type NPH3 and constitutive membrane-bound / unbound variants.

R14: Thanks for pointing this out! We now (Fig. 7e) separated the 3 panels and listed the appropriate NPH3 variants at the bottom of the individual panels.

Q15: Line 397: 'also in plants' is not necessary (any more than also in bacteria, also in animals etc)!

R15: We deleted it (line 427).

With best regards, also on behalf of my co-authors,
Claudia Oecking

REVIEWERS' COMMENTS

Reviewer #1 (Remarks to the Author):

I would like to thank the Authors for providing answers to my questions. I appreciate the time and effort they have taken to improve the manuscript. I admire the comprehensive attitude to the science presented in this study. I fully recommend this paper for publishing in Nature Communications.

If possible, I would like to ask for small additions during the manuscript processing:

Lines 42-45, please add the abbreviated names of proteins.

Fig.4F Please show also the green channel separately.

M&M Please provide the Catalog Numbers for antibodies.

Reviewer #2 (Remarks to the Author):

The authors have taken my comments on board and I am pleased the new format of the MS. I would therefore like to recommend this MS for publication.

Apologies for not noticing it in the previous MS, but I just spotted something in Figure 7C. GFP-NPH3-4K/A does unexpectedly show some phosphorylation of S744, despite it being located in the cytoplasm. I guess that PM localised phot1 still is still able to phosphorylate a small proportion of cytosolic NPH3, thus explaining the partial phosphorylation. The fact that GFP-NPH3-4K/A is phosphorylated and cytoplasmic and yet still not functional actually lends even more weight to the authors hypothesis that it is the cycling is of NPH3 is essential for its function. Please note that I'm only raising this observation because I have the opportunity to do so. The authors shouldn't feel at all obliged to discuss it in the MS if they feel that it is not warranted!

Again, congratulations on the MS, really nice work!

Best wishes,

Scott Hayes